# Identification and Characterization of the *BBX* Gene Family in *Bambusa pervariabilis* × *Dendrocalamopsis grandis* and Their Potential Role under Adverse Environmental Stresses

**DOI:** 10.3390/ijms241713465

**Published:** 2023-08-30

**Authors:** Yi Liu, Yaxuan Wang, Jiao Liao, Qian Chen, Wentao Jin, Shuying Li, Tianhui Zhu, Shujiang Li

**Affiliations:** 1College of Forestry, Sichuan Agricultural University, Chengdu 611130, China; liuyi20186021@163.com (Y.L.); xx18996806292@163.com (Y.W.); liaojiao0421@163.com (J.L.); 18782853057@163.com (Q.C.); 18193386535@163.com (W.J.); leesy0710@126.com (S.L.); 10627@sicau.edu.cn (T.Z.); 2National Forestry and Grassland Administration Key Laboratory of Forest Resources Conservation and Ecological Safety on the Upper Reaches of the Yangtze River, Chengdu 611130, China

**Keywords:** B-box gene, *Bambusa pervariabilis × Dendrocalamopsis grandis*, bioinformatics analyses, abiotic stress, biotic stress

## Abstract

Zinc finger protein (ZFP) transcription factors play a pivotal role in regulating plant growth, development, and response to biotic and abiotic stresses. Although extensively characterized in model organisms, these genes have yet to be reported in bamboo plants, and their expression information is lacking. Therefore, we identified 21 B-box (*BBX*) genes from a transcriptome analysis of *Bambusa pervariabilis × Dendrocalamopsis grandis*. Consequently, multiple sequence alignments and an analysis of conserved motifs showed that they all had highly similar structures. The *BBX* genes were divided into four subgroups according to their phylogenetic relationships and conserved domains. A GO analysis predicted multiple functions of the *BBX* genes in photomorphogenesis, metabolic processes, and biological regulation. We assessed the expression profiles of 21 *BBX* genes via qRT-PCR under different adversity conditions. Among them, eight genes were significantly up-regulated under water deficit stress (*BBX4*, *BBX10*, *BBX11*, *BBX14*, *BBX15*, *BBX16*, *BBX17*, and *BBX21*), nine under salt stress (*BBX2*, *BBX3*, *BBX7*, *BBX9*, *BBX10*, *BBX12*, *BBX15*, *BBX16*, and *BBX21*), twelve under cold stress (*BBX1*, *BBX2*, *BBX4*, *BBX7*, *BBX10*, *BBX12*, *BBX14*, *BBX15*, *BBX17*, *BBX18*, *BBX19*, and *BBX21*), and twelve under pathogen infestation stress (*BBX1*, *BBX2*, *BBX4*, *BBX7*, *BBX10*, *BBX12*, *BBX14*, *BBX15*, *BBX17*, *BBX18*, *BBX19*, and *BBX21*). Three genes (*BBX10*, *BBX15*, and *BBX21*) were significantly up-regulated under both biotic and abiotic stresses. These results suggest that the *BBX* gene family is integral to plant growth, development, and response to multivariate stresses. In conclusion, we have comprehensively analyzed the *BDBBX* genes under various adversity stress conditions, thus providing valuable information for further functional studies of this gene family.

## 1. Introduction

*Bambusa pervariabilis* × *Dendrocalamopsis grandis* (hybrid bamboo) is a highly valued new bamboo variety, which has been successfully cultivated by using *Bambusa pervariabilis* as the female parent and *Dendrocalamopsis grandis* as the male parent. It is widely recognized as a leading economic bamboo variety in the Yangtze River Basin of China, owing to its large and multiple shoots, high yield, good edibility, high branching, and excellent papermaking properties [1,2]. However, significant dieback of hybrid bamboo in cultivated areas has occurred as a result of *Arthrinium phaeospermum* (Corda) M.B. Ellis’s shoot blight’s recent introduction, spanning over 3000 hm^2^. Previous studieshave shown that the main pathogen causing shoot blight in hybrid bamboo is *Arthrinium phaeospermum* [3]. The disease has spread rapidly, causing extensive ecological and economic losses by destroying the ecological barriers of the Yangtze River Basin [4,5]. Typical symptoms include brown tongue-shaped spots at the fork of the bamboo node, particularly on new shoots in early spring (4–8 °C). The condition can worsen the quality of bamboo timber in mild cases or lead to the total loss of the bamboo stand in severe cases [5,6]. While research into the disease’s physiology has been conducted, molecular biology studies have also progressed [7]. To combat agricultural diseases, engineering techniques are rapidly being used to create disease-resistant cultivars, with resistance genes now regarded as the most efficient and affordable option available. However, such research on bamboo plants, particularly *B. pervariabilis* × *D. grandis*, has been limited. Previous studies have demonstrated that the expression levels of numerous genes implicated in biological processes, such as signal transduction, cell wall composition synthesis, secondary metabolite synthesis, and redox reaction, fluctuate significantly following hybrid bamboo infection by pathogens. Our team has identified 21 candidate genes that may be responsible for the resistance to hybrid bamboo shoot blight, such as *CCoAOMT2*, *CAD5*, *BBX*, *gdsl-II*, and *Myb4l* [3,5,7]. However, low temperatures and water deficits foster the hybrid bamboo shoot blight disease, providing favorable conditions for *A. phaeospermum* infestation [2]. Consequently, the need to explore broad-spectrum stress resistance genes in hybrid bamboo is urgent.

About 15% of all transcription factors in plants belong to the family of zinc finger transcription factors, which are important for plant growth and development [8]. These proteins contain zinc finger structural domains that are stabilized by metal ions, including zinc, and have distinct characteristics for DNA, RNA, or protein interactions [9,10], influencing plant responses to changing environmental conditions [11]. Among the zinc finger proteins, a functionally diverse subgroup known as B-box zinc proteins (BBX) plays a vital role. The *BBX* family is highly conserved in all multicellular species, including blue-green algae and mosses, and consists of one or two BBX motifs that are predicted to participate in protein–protein interactions [12,13,14,15]. Besides the conserved BBX structural domains, some BBX proteins possess family-specific structural domains, such as the CCT domain. Based on the number of BBX or CCT structural domains they contain, the *BBX* gene family can be categorized into five subfamilies [10].

Initially, the *BBX* genes in *Arabidopsis thaliana* were labelled Class *CO* (*COL*) based on the first *BBX* gene identified in *Arabidopsis thaliana*, *CO* (CONSTANS) [16]. The subsequent genomic identification of all 32 *BBX* members in *Arabidopsis thaliana* led to a unified nomenclature, and the *AtBBX* gene was introduced [10]. Previous studies have reported that BBX proteins in *Arabidopsis thaliana* serve as key regulators of plant growth and development processes, such as seedling photomorphogenesis, photoperiodic regulation of flowering, shade avoidance, and responses to biotic and abiotic stresses [17,18]. Further research has shown that *BBX* transcription factors play a vital role in various life activities, including photomorphogenesis [19], flowering [20], shade avoidance responses [21], biotic and abiotic stress responses [22,23], and phytohormone signaling across different plant species [24]. Additionally, BBX proteins have been linked to DNA binding or protein interactions [25]. In certain plants, B-box zinc finger proteins have been observed to induce immune responses of related genes in response to external abiotic stresses—for instance, *Oryza sativa* in response to drought, high amounts of salt, low-temperature, abscisic acid stress [26]; *Petunia hybrida* in response to low-temperature stress [27]; *Solanum lycopersicum* in response to drought stress [14]; and *Brassica napus* in response to various abiotic stresses (heat, cold, salt, and drought) or hormones (abscisic acid, methyl jasmonate, and gibberellin) [28]. Moreover, B-box zinc finger proteins have also been found to impact plant growth and development significantly [29,30]. The plant genebank database has identified 32, 30, and 29 *BBX* gene families in *Arabidopsis thaliana* [10], *Oryza sativa* [26], and *Solanum lycopersicum* [14], respectively. Based on this information, we hypothesize that the *BBX* gene in hybrid bamboo confers broad-spectrum resistance against abiotic and biotic stresses.

Transcriptomics has become a widespread method for screening plants for disease-resistant compounds and identifying key differentially expressed genes in recent years [5]. Our research group conducted four different treatments on three *B. pervariabilis* × *D. grandis* varieties (#3: medium level of resistance, #6: highly resistant, and #8: susceptible): (1) the AP toxin treatment alone (which induced disease resistance); (2) pathogen spore suspension inoculation after the AP toxin treatment (induction followed by pathogen infection); (3) the sterile water treatment alone (the negative control); and (4) the pathogen spore treatment alone (the positive control) [4]. Transcriptome sequencing was performed on all treatment groups. This study used bioinformatics methods to identify *BBX* gene family members from the RNA-seq database. It analyzed their expression under various adverse stress conditions, such as water deficits, high amounts of salt, cold temperatures, and pathogens, thus providing a theoretical foundation for future in-depth studies on the function of the *BBX* gene in *B. pervariabilis* × *D. grandis*.

## 2. Results

### 2.1. Identification and Analysis of the BBX Gene Family

As seed sequences, we included all plant BBX sequences and conducted a local blast search of the transcriptome database. A data integration analysis allowed us to obtain a total of 43 *BBX* sequences by eliminating redundant sequences. Using SMART (v8) and Pfam (27.0) online software, we identified the conserved structural domains, culminating in 21 BBX protein sequences. Subsequently, we obtained the *BBX* sequence and organized the sequence under the *BBX* numbering order (*BDBBX1-21*) of *B. pervariabilis* × *D. grandis* in compliance with the unigene ID numbers in the transcriptome (Table 1).

The protein sequences of the *BDBBX* gene family ranged from 83 to 409 aa in length, with corresponding relative molecular weights varying from 8.413 to 44.319 kD. The isoelectric points (pI) ranged from 4.69 to 11.9. Among the 21 *BDBBX* genes, 11 had isoelectric points greater than 7, making them basic and indicating that these genes primarily functioned in subcellular alkaline environments. The other 10 genes had isoelectric points less than 7, rendering them acidic and signifying that they functioned primarily in acidic subcellular environments. A further analysis revealed that only one BDBBX protein had an instability index of less than 40, making it a stable protein, while the remaining 20 BDBBX proteins were unstable. The aliphatic amino acid index ranged from 41.08 to 90.29, which correlated with the thermal stability of the proteins and indicated differences in the thermal stability among the BDBBX family proteins. All BDBBX family proteins have a hydrophobicity index of <0, demonstrating their hydrophilic nature. Concerning the subcellular localization of BBX proteins, all but BDBBX7 and BDBBX20 were found to be localized in the nucleus, with BDBBX6, BDBBX9, BDBBX19, and BDBBX21 appearing in the chloroplast, cell membrane, and mitochondrial sites, in addition to the nucleus. These observations are consistent with the finding that transcription factors localize in the nucleus. Using SignalP 4.1 server software to predict the presence of signal peptides, we found only one BDBBX family protein (BDBBX6) containing signal peptides, which amounted to 3.6% of the total gene family. This study highlights the marked differences in the amino acid sequence length and protein properties of BBX proteins, implying varied biological properties.

### 2.2. Conserved Motif Analysis of the BBX Gene Family

To analyze the conserved structural domains of the BBX family, we utilized the online software Pfam (27.0), as well as CD-Search in Blast (Figure 1A). Among the 21 BDBBX proteins, only BDBBX6, BDBBX17, and BDBBX19 contained three structural domains simultaneously, accounting for 14.3% of the total number of genes. Of these, six BBX proteins, namely BDBBX2, BDBBX7, BDBBX10, BDBBX12, BDBBX14, and BDBBX21, had one Bbox1_BBX-like domain and one Bbox2_BBX-like domain, whereas four BBX proteins, BDBBX13, BDBBX16, BDBBX18, and BDBBX20, had one Bbox1_BBX-like domain and one CCT domain. On the other hand, the remaining eight BDBBX proteins contained only one Bbox1_BBX-like domain.

We analyzed the predicted conserved motifs in the BBX protein using the online software MEME (4.12.0) database, and the results revealed that the predicted BBX family comprised 10 conserved motifs (Figure 1B), with differences in type and number observed among the BBX family members. A further functional annotation of the 10 conserved motifs using the online Pfam (27.0) software and SMART (v8) software (Table 2) showed that motifs 1, 4, and 7 were B_box and motif 2 was CCT, while the remaining six motifs were unannotated, with unknown functions. The presence of motif 1 across almost all BDBBX proteins indicates that it is the most conserved motif in the family, being the basic motif that constitutes the family. Seven BBX proteins, namely BDBBX6, BDBBX13, BDBBX16, BDBBX17, BDBBX18, BDBBX19, and BDBBX20, contained motif 2, whereas BDBBX2, BDBBX9, BDBBX10, BDBBX14, BDBBX15, and BDBBX21 were the six BBX proteins containing motif 4. Moreover, only BDBBX6 and BDBBX19 held motif 7. The presence of different motifs among different BDBBX proteins suggests that BBX proteins may have different biochemical characteristics and biological functions. Furthermore, Figure 1B shows that the conserved motifs Motif2, Motif7, and Motif8 are all highly conserved, and analyzing these motifs will be beneficial in exploring the structural composition of the protein.

### 2.3. Phylogenetic Analysis of the BBX Gene Family

To investigate the phylogenetic relationships within the *BBX* gene family of *B. pervariabilis* × *D. grandis*, a phylogenetic tree was constructed using MEGA (11.0.13) software that included *BBX* gene family members from *B. pervariabilis* × *D. grandis*, *Arabidopsis thaliana*, and *Oryza sativa* (Figure 1C). Based on the phylogenetic branching, the results demonstrated that the 21 BDBBX proteins could be categorized into four subfamilies. Group V was identified as the largest subfamily, which contained eight BDBBX proteins, representing 38.11% of the total number of gene families. Group I, Group III, and Group IV contained three, four, and six BDBBX proteins, respectively. The evolutionary tree illustrated that BDBBX was more closely related to OsBBX and distantly related to AtBBX. Within this, OsBBX1 and BDBBX2, OsBBX24 and BDBBX4, OsBBX30 and BDBBX7, OsBBX6 and BDBBX9, OsBBX9 and BDBBX13, OsBBX4 and BDBBX14, and OsBBX11 and BDBBX21 were identified as direct homologous genes. These BBX proteins are presumed to exhibit similar biological functions to their rice counterparts.

### 2.4. Structural Information Analysis of the BBX Gene Family

A protein secondary structure prediction for the members of the *BBX* gene family was made using the online SOPMA program. The results indicated that all BBX proteins consisted of four secondary structures: alpha helix, beta-turned helix, random coil, and extended strand. Of these, the primary and secondary structure predicted for 17 BBX proteins was the random coil, accounting for 80.95% of the total, whereas the remaining 4 proteins had predominantly alpha helix structures. The secondary structure predictions for 19 proteins indicated a low number of beta-turned helix structures, accounting for 90.48%, while the remaining two proteins had the lowest number of predicted secondary structures in the form of extended strands. Furthermore, a majority of the proteins exhibited α-helix and irregular curl structures as the main components of their secondary structure. Therefore, it can be concluded that α-helix and irregularly coiled structures are the primary components of the BBX protein, while β-turned and extended strand structures play a secondary modifying role, scattered throughout the protein sequence.

Upon conducting homology modelling of the 21 BBX proteins (Figure 2), it was observed that the BDBBX proteins within each subfamily exhibited highly conserved structural features, while the proteins between subfamilies displayed significant differences in their spatial structures. Based on these results, it was postulated that BDBBX proteins within the same subfamily may have analogous functions, whereas BDBBX proteins between different subfamilies may differ in the functions they perform.

### 2.5. GO Analysis of the BBX Gene Family

The distribution of the GO levels is demonstrated in Figure 3A. Among the 21 *BDBBX* genes, four genes (*BDBBX7*, *BDBBX10*, *BDBBX14*, and *BDBBX15*) were functional in all three categories, while 11 genes displayed functionalities in two categories. Of these, nine genes exhibited functionalities in cellular composition (CC) and molecular function (MF), while two genes were functional in cellular composition (CC) and biological process (BP). In terms of molecular function (MF), three genes possessed unique and specific functions. Furthermore, three genes lacked GO functional annotations.

In the GO enrichment analysis (Figure 3B), a total of 29 GO terms were identified. Among these, 21 GO terms were linked to biological processes (BP), including metabolic processes, cellular processes, biological regulation, and responses to stimuli, implying that *BBX* genes play important roles in biotic and abiotic stresses, growth and development, and biochemical processes. Additionally, five GO terms were associated with cellular components (CCs), including organelles, cells, and cell parts, suggesting that they are linked to the regulation of transcriptional machinery. There were also three GO terms related to molecular functions (MFs), such as binding function and transcriptional regulatory activity.

### 2.6. Potential Phosphorylation Sites and Glycosylation Analysis of the BBX Gene Family

The online software NetNGlyc-1.0 was utilized to predict phosphorylation sites, and the results revealed that a total of three protein kinase phosphorylation sites, including serine, threonine, and tyrosine, were predicted to be phosphorylated. BDBBX1, BDBBX13, and BDBBX17 were predicted to undergo phosphorylation modifications at two protein kinase phosphorylation sites, namely serine and threonine, respectively. BDBBX3 and BDBBX7 were predicted to undergo phosphorylation modifications at two protein kinase phosphorylation sites: serine and tyrosine. Serine, threonine, and tyrosine triple phosphorylation sites were identified in the remaining *BDBBX* genes (Appendix A, Table 3).

The online software NetPhos 3.1 was used for glycosylation site prediction, and the results indicated that eight proteins in the BDBBX family, namely BDBBX2, BDBBX4, BDBBX9, BDBBX10, BDBBX14, BDBBX15, BDBBX17, and BDBBX20, possessed potential glycosylation sites (Table 3). Among these proteins, BDBBX2 and BDBBX20 were anticipated to have two potential glycosylation sites, while BDBBX14 displayed the highest probability of containing a glycosylation site.

### 2.7. Protein–Protein Interaction Networks

Differential gene protein interaction networks were analyzed using interactions gathered from the STRING protein interaction database. A homology analysis of AtBBX was utilized to forecast the interaction network of 21 candidate BDBBXs. The results revealed that 14 proteins in the BDBBX family were predicted to interact (Figure 4). Among them, the BDBBX8 protein demonstrated the highest connectivity, interacting with nine BDBBX proteins from the hybrid bamboo.

Expanding the interaction network of other proteins with BBX proteins revealed that BDBBX10, BDBBX11, and BDBBX21 are interactive with SIGE proteins, which play a crucial role in blue-light-mediated transcription and the reproduction of psbD, indicating that BDBBX10, BDBBX11, and BDBBX21 proteins might be involved in this regulation. Moreover, BDBBX8, BDBBX10, BDBBX11, BDBBX13, BDBBX19, and BDBBX21 interact with CCA1 proteins to regulate plant circadian rhythms cooperatively. Furthermore, BDBBX4, BDBBX9, BDBBX10, BDBBX11, BDBBX12, BDBBX15, and BDBBX21 are interactive with the ubiquitin ligase COP1 protein and contribute to photomorphogenesis. Finally, BDBBX11 and BDBBX21 were found to interact with the AT5G64170 protein, which may suggest their involvement in translocation.

### 2.8. Inducible Expression Analysis of the BBX Gene in Bambusa pervariabilis × Dendrocalamopsis grandis under Abiotic Stress

A study was conducted to examine how water deficit stress affected the expression of 21 *BBX* genes in three hybrid bamboo varieties (Figure 5). The results showed that for variety 3, the peak expression levels of *BBX6*, *BBX10*, *BBX11*, *BBX14*, *BBX15*, *BBX17*, and *BBX21* were observed at one of the 6-h, 12-h, and 24-h periods. Meanwhile, *BBX4* and *BBX16* exhibited two peaks at 6 h and 24 h, respectively. The expression levels of all these 9 *BBX* genes were up-regulated compared to the control group, indicating that they may confer water deficit tolerance. Similarly, in variety 6, the expression levels of *BBX6*, *BBX10*, *BBX11*, *BBX14*, *BBX15*, *BBX16*, *BBX17*, and *BBX21* exhibited peak expression levels at one of the 6-h, 12-h, or 24-h periods, while *BBX2*, *BBX3*, and *BBX4* exhibited two peaks at 6 h and 24 h, respectively. These 11 *BBX* genes were also up-regulated compared to the control group, indicating their potential role in conferring water deficit tolerance. In variety 8, the expression levels of *BBX7*, *BBX10*, *BBX14*, *BBX15*, *BBX16*, *BBX17*, and *BBX21* exhibited peak expression levels at one of the 6-h, 12-h, or 24-h periods, while *BBX4* and *BBX11* showed two peaks at 6 h and 24 h, respectively. Similarly, all 9 *BBX* genes in this variety were up-regulated, suggesting their possible contribution to water deficit tolerance. Furthermore, interspecific comparisons indicated that the expression levels of eight genes (*BBX4*, *BBX10*, *BBX11*, *BBX14*, *BBX15*, *BBX16*, *BBX17*, and *BBX21*) were consistently higher in all three varieties of hybrid bamboo compared to the control group, suggesting their potential role as genes involved in water deficit tolerance mechanisms in hybrid bamboo.

An analysis of changes in the expression of 21 *BBX* genes of a single variety under salt stress conditions was performed (Figure 6). The results demonstrated that in variety 3, the peak expression levels of *BBX7*, *BBX10*, *BBX12*, *BBX15*, and *BBX21* were observed at one of the 6-h, 12-h, and 24-h periods, while *BBX2*, *BBX3*, *BBX9*, and *BBX16* exhibited peaks at 6 h and 24 h, respectively. All nine *BBX* genes showed up-regulated expression levels and were deemed to be salt tolerant. Similarly, in variety 6, the peak expression levels of *BBX2*, *BBX7*, *BBX9*, *BBX10*, *BBX12*, *BBX16*, and *BBX21* were observed at one of the 6-h, 12-h, and 24-h periods, while *BBX3* and *BBX15* exhibited two peaks at 6 h and 24 h, respectively. The expression levels of all nine *BBX* genes were up-regulated and were presumed to be salt tolerant. Additionally, in variety 8, the peak expression levels of *BBX2*, *BBX7*, *BBX9*, *BBX13*, *BBX16*, and *BBX21* were observed at one of the 6-h, 12-h, and 24-h periods, while *BBX3*, *BBX10*, *BBX12*, *BBX14*, and *BBX15* exhibited two peaks at 6 h and 24 h, respectively. All 11 *BBX* genes showed an up-regulated expression trend and were deemed to have salt tolerance potential. Furthermore, interspecific comparisons revealed that the expression levels of nine genes (*BBX2*, *BBX3*, *BBX7*, *BBX9*, *BBX10*, *BBX12*, *BBX15*, *BBX16*, and *BBX21*) were consistently higher in all three varieties of hybrid bamboo compared with the control group, showing an up-regulated trend of gene expression, hinting that they may be salt-tolerant genes in hybrid bamboo.

An analysis of changes in the expression of 21 *BBX* genes of a single variety under cold stress conditions was conducted (Figure 7). Our findings showed that for variety 3, the expression levels of *BBX1*, *BBX2*, *BBX4*, *BBX7*, *BBX12*, *BBX14*, *BBX17*, *BBX18*, *BBX19*, and *BBX21* exhibited peaks at one of the 6-h, 12-h, and 24-h periods, while *BBX8*, *BBX10*, and *BBX15* exhibited two peaks at 6 h and 24 h, respectively. All 13 *BBX* genes demonstrated up-regulated expression levels and were deemed to be resistant to cold stress. Similarly, in variety 6, the peak expression levels of *BBX1*, *BBX2*, *BBX3*, *BBX4*, *BBX7*, *BBX9*, *BBX10*, *BBX12*, *BBX15*, *BBX16*, and *BBX18* were observed at one of the 6-h, 12-h, and 24-h periods, while *BBX6*, *BBX11*, *BBX14*, *BBX17*, *BBX19*, and *BBX21* exhibited two peaks at 6 h and 24 h, respectively. All 17 *BBX* genes exhibited up-regulated expression levels and were deemed to be resistant to cold stress. Additionally, in variety 8, the peak expression levels of *BBX1*, *BBX4*, *BBX7*, *BBX10*, *BBX11*, *BBX12*, *BBX14*, *BBX15*, *BBX17*, *BBX18*, *BBX19*, and *BBX21* were observed at one of the 6-h, 12-h, and 24-h periods, while only *BBX2* exhibited two peaks at 6 h and 24 h, respectively. All 13 *BBX* genes exhibited up-regulated expression levels and were deemed to be resistant to cold stress. Further interspecific comparisons indicated that the expression levels of a total of 12 genes (*BBX1*, *BBX2*, *BBX4*, *BBX7*, *BBX10*, *BBX12*, *BBX14*, *BBX15*, *BBX17*, *BBX18*, *BBX19*, and *BBX21*) were consistently higher in all three varieties of hybrid bamboo than in the control group, exhibiting an up-regulation trend of gene expression. The expression levels of these genes in all three bamboo hybrids were higher compared to those of the control group.

### 2.9. Induced Expression Analysis of the BBX Gene in Bambusa pervariabilis × Dendrocalamopsis grandis under Disease Infestation

We used qRT-PCR testing to check the transcript levels of the *BBX* gene in *B. pervariabilis* × *D. grandis* bamboo while it was being infested by *A. phaeospermum*. Our findings reveal that the expression of 21 *BBX* genes in three varieties varied across different periods (Figure 8). In variety 3, we found that the *BBX1*, *BBX2*, *BBX4*, *BBX10*, *BBX14*, *BBX15*, *BBX18*, and *BBX19* expression levels peaked at one of the periods of 3 d, 5 d, 7 d, and 10 d. Additionally, the *BBX7*, *BBX8*, *BBX11*, *BBX12*, *BBX17*, and *BBX21* genes’ expression levels had two peaks at 5 d and 10 d, respectively. We observed that all 14 up-regulated *BBX* genes exhibited resistance against *A. phaeospermum* infection. Similarly, in variety 6, we detected that the expression levels of *BBX1*, *BBX2*, *BBX5*, *BBX6*, *BBX7*, *BBX8*, *BBX9*, *BBX10*, *BBX11*, *BBX12*, *BBX14*, *BBX15*, *BBX17*, *BBX18*, and *BBX19* peak at 3 d, 5 d, 7 d, and 10 d, whereas *BBX4* and *BBX21* had two peaks at 5 d and 10 d, respectively. All 17 up-regulated *BBX* genes showed resistance to *A. phaeospermum*. Further, in variety 8, we found that the expression of *BBX1*, *BBX2*, *BBX4*, *BBX14*, *BBX15*, *BBX17*, *BBX18*, and *BBX19* showed peaks at 3 d, 5 d, 7 d, or 10 d, while *BBX7*, *BBX10*, *BBX12*, and *BBX21* had two peaks at 5 d and 10 d, respectively. All 12 up-regulated *BBX* genes were resistant to *A. phaeospermum* infection. The results of the disease index and incidence rate showed that (Appendix A) among the three hybrid bamboo varieties, the disease index and incidence rate ranged from low to high by the pattern of high resistance 6# to medium resistance 3# to susceptible 8#. The disease index of high resistance 6# was the lowest at 8.33, and the incidence rate was 16.67%. The susceptible 8# had a higher disease index of 27.78 and an incidence of 50.00%. Interestingly, we observed that the proportion of down-regulated genes in the three varieties correlated with their degree of resistance: five (highly resistant #6), seven (moderately resistant #3), and eight (susceptible #8) down-regulated genes. Meanwhile, among the genes with up-regulated gene expression levels, it was found that the high and low gene expression levels coincided with the pattern of high-resistance varieties being higher than medium-resistance varieties which were higher than susceptible varieties. For example, the expression level of *BBX10* was fifteen times higher than the control level in variety 6, while the expression level of this gene was ten times and five times higher than the control level in varieties 3 and 8, respectively. The expression level of *BBX15* was 25 times higher than the control level in variety 6, while the expression level of this gene was 18 times and 9 times higher than the control level in varieties 3 and 8, respectively. These results indicate that the increase in the expression level of the *BBX* gene is positively correlated with the resistance of the varieties. Notably, 12 genes, namely *BBX1*, *BBX2*, *BBX4*, *BBX7*, *BBX10*, *BBX12*, *BBX14*, *BBX15*, *BBX17*, *BBX18*, *BBX19*, and *BBX21*, showed a trend of the up-regulation of the expression level in all the three varieties as compared to the control. These genes are presumably the most probable candidates for disease resistance in the hybrids. In addition, the results of Appendix A showed that 12 disease resistance candidate genes were negatively correlated with the incidence rate, among which the gene expression levels of *BBX10* and *BBX21* were significantly negatively correlated with the incidence rate.

### 2.10. Comparison of Amino Acid Sequences of BBX Proteins of Different Species

The previous experiments indicated that three BDBBX proteins in hybrid bamboo demonstrated resistance against different biotic and abiotic stresses. These proteins, namely BDBBX10, BDBBX15, and BDBBX21, were chosen for multiple sequence alignment, which also included three species of *Setaria italica*, *Oryza sativa*, and *Panicum virgatum* (Figure 9). The outcome revealed that BDBBX10 exhibited 83.33%, 81.60%, and 78.00% similarities to SiBBX22, OsBBX22, and PvBBX22, respectively, with SiBBX22 having the highest degree of similarity. The similarities of BDBBX15 with the three Gramineae species compared exceeded 65.17%. BDBBX21 shared the highest similarity of 81.01% with OsBBX24 as well as 78.93% and 77.91% with SiBBX22 and PvBBX22, respectively. In conclusion, BDBBX10, BDBBX15, and BDBBX21 demonstrated substantial homology with the amino acid sequences of *Setaria italica*, *Oryza sativa*, and *Panicum virgatum*. Notably, BDBBX10 and BDBBX21 showed a higher degree of similarity to these Gramineae crops, with BDBBX10 sharing the highest degree of similarity to *Panicum virgatum* and BDBBX21 sharing the highest degree of similarity with *Oryza sativa*. Furthermore, we observed significant similarities between sequences from the four plants in the range of 1–50 amino acids, exceeding 95%.

## 3. Discussion

*B. pervariabilis* × *D. grandis* has gained immense popularity due to its easy propagation, adaptability, and strong resistance, making it a widely promoted economic timber species in China’s Grain for Green Project [7]. Given its high economic, ecological, and social benefits, it is imperative to explore the resistance mechanism of *B. pervariabilis* × *D. grandis* comprehensively to ensure its full conservation and utilization. Furthermore, such studies will unveil its immense development potential and application prospects. Plant transcription factors are pivotal in regulating gene expression by interacting with DNA and other proteins through their specific structural domains. Among them, *BBX* transcription factors are crucial in regulating various life activities. Extensive research has been carried out on their structural attributes and biological functions in the model plants *Arabidopsis thaliana*, *Oryza sativa*, and *Populus* [31,32]. However, studies on the composition and function of *BBX* family members of *B. pervariabilis* × *D. grandis* are scant due to the current lack of genomic libraries and related annotation information. Identifying the *BBX* gene family of *B. pervariabilis* × *D. grandis* based on histological developments is crucial to understanding the mechanisms of gene expression and regulation in response to plant hormones, biotic and abiotic adversity stresses, and plant growth and development.

The *BBX* gene family has gained significant attention from the scientific community recently. Genome-wide identification analyses of *BBX* genes have been conducted in plants such as *Oryza sativa* [26], *Arabidopsis thaliana* [10], *Solanum lycopersicum*, *Pyrus*, and *Solanum tuberosum* [14,33,34]. In this study, we conducted a comprehensive analysis and identification of the *BDBBX* gene family using a bioinformatics approach based on the transcriptomic data of *B. pervariabilis* × *D. grandis* under various treatment conditions, with information from *Arabidopsis thaliana* and *Oryza sativa BBX* family genes. We identified a total of 21 *BBX* gene family members in *B. pervariabilis* × *D. grandis*, and the *BBX* gene family was classified and named in detail through a phylogenetic analysis. The number of *BBX* gene family members is relatively consistent across crops, with 30, 32, 29, and 30 *BBX* gene members identified in *Oryza sativa*, *Arabidopsis thaliana*, *Solanum lycopersicum*, and *Solanum tuberosum*, respectively [10,14,26,34]. The number of *BBX* genes in *B. pervariabilis* × *D. grandis* is lower, with only 21 *BBX* genes compared to homologous genes in *Solanum lycopersicum* (29) [14], *Arabidopsis thaliana* (32) [29], and *Oryza sativa* (30) [23]. However, two species from the Gramineae family also contain fewer *BBX* members, with 22 from stiff brome and 19 from millet [35]. This variation may exist because the genomes of these three species have not been entirely sequenced or because they have compact, straightforward genomes. Although the quantitative differences were not significant, the types of *BBX* genes differed between species. For instance, in *Solanum lycopersicum*, there were eight, five, ten, and six *BBX* gene members in two tandem B-BOX plus CCT structural domains, one B-BOX plus CCT structural domain, two tandem B-BOX structural domains, and one B-BOX structural domain, respectively [14]. In *Arabidopsis thaliana*, the corresponding numbers were thirteen, four, eight, and seven [10], while in *B. pervariabilis* × *D. grandis*, they were three, four, six, and eight. These findings imply that the *BBX* gene may have originated from a common ancestor in several species and independently spread following the divergence of dicotyledons and monocotyledons [36].

To better understand the evolutionary history of the *BBX* gene family, we constructed a phylogenetic tree of *BBX* genes from monocots (*B. pervariabilis* × *D. grandis* and *Oryza sativa*) and dicots (*Arabidopsis thaliana*). Previous studies have reported that *Arabidopsis thaliana* has 32 members [10] while *Oryza sativa* has 30 members in the *BBX* gene family [23], with both divided into five subfamilies. Interestingly, the *BBX* gene subfamily distribution in *B. pervariabilis* × *D. grandis* is similar to that of *Arabidopsis thaliana* and divided into four distinct subfamilies. Subfamily I, which includes six members of the *Arabidopsis* family (*AtBBX1-AtBBX6*), has been implicated in flowering regulation, stomatal opening, and stress responses [18]. We hypothesize that *BDBBX6*, *BDBBX17*, and *BDBBX19* of the same subfamily may have similar functions. Meanwhile, four members of the *Arabidopsis* family classified as subfamily IV (*AtBBX22-AtBBX25*) have been reported to be involved in light morphogenesis [18]. We suggest that *BDBBX2*, *BDBBX7*, *BDBBX10*, *BDBBX12*, *BDBBX14*, and *BDBBX21* of the same subfamily are also likely to be involved in the regulation of light morphogenesis in *B. pervariabilis* × *D. grandis*. A further phylogenetic analysis indicates that the evolutionary relationships of *BBX* genes in *B. pervariabilis* × *D. grandis* are closer to *Oryza sativa*, with seven pairs of direct homologous genes showing up to 100% homology. Based on the evolutionary trajectory of the BBX structural domain in green plants, it appears that the original BBX protein had only one BBX structural region, which underwent a duplication event in later evolution, resulting in the formation of the CCT domain. Its subsequent evolution added different structural types to the BBX protein, including the deletion of 2 BBX and 1 CCT structural domains, as well as further duplication events of the BBX structural domain [13]. This diversity in the BBX protein’s structural domains is reflected in the 21 *BDBBX* genes, in which 12 have only one B-box 1 domain due to deletions during evolution. The CCT structural domain has been reported to be highly conserved, while a detailed sequence comparison of the B-box1 and B-box2 structural domains of *Oryza sativa* reveals that the B-box1 structural domain is highly conserved compared to B-box2 [37]. It is hypothesized that a deletion process occurred in the B-box2 domain, giving rise to the B-box1 domain [35]. This may explain why *AtBBX18-21* in subfamily V contains both B-box1 and B-box2 domains, while homologous *BDBBX* genes such as *BDBBX1*, *BDBBX3*, *BDBBX9*, *BDBBX11*, and *BDBBX15* contain only one B-box1 domain.

The mechanisms of tandem and mass duplication are essential for multiplying members of gene families in the genome throughout evolution [38]. Large-scale duplication events have been reported for the *SPL* gene family of *Phyllostachys edulis*, and studies of the *BBX* gene family have yielded similar results [39]. In the *BDBBX* gene family, segmental duplications such as *BDBBX7* and *BDBBX12* as well as *BDBBX6* and *BDBBX19* have also been observed. These findings suggest that segmental duplication events contribute to the expression of the *BBX* gene family in *B. pervariabilis* × *D. grandis*.

The GO database is a widely used tool for standardizing the functional classification of genes in transcriptome analyses [36]. Studies have shown that members of the *BDBBX* gene family exhibit different functions. Through a GO enrichment analysis, we found that the molecular functions of *BDBBX* genes are mainly enriched in zinc ion binding (GO:0008270), DNA binding (GO:0003677), and transcription factor activity (GO:0003700). The cellular components are mostly enriched in cytoplasmic vesicles (GO:0016023), mitochondria (GO:0005739), and intracellular regions (GO:0005622). In terms of biological processes, the genes are enriched in signal transduction (GO:0007165), photomorphogenesis (GO:0009640), and response to red or far-red light (GO:0009639). Moreover, *BBX* genes have similar functions in *Arabidopsis thaliana* [10] and *Oryza sativa* [26], indicating that *BBX* transcription factors can perform similar roles across species.

The B-box domain plays a crucial role in transcriptional regulation in plants by forming heterodimers within the BBX protein family or with other proteins [40]. Therefore, studying protein–protein interaction networks is essential for understanding the function of BDBBX proteins and maintaining complex biological network systems [41]. In *Arabidopsis*, AtBBX21 has been shown to form heterodimers with HY5 and ABI5, respectively, to regulate light-mediated ABA signaling and influence photomorphogenesis [42]. Moreover, BBX proteins from different plants may exhibit cross-species effects. For example, the N-terminal BBX region of AtBBX32 interacts with the BBX protein of GmBBX62 in *Glycine max* [43]. Our protein interaction network analysis showed that 14 BDBBX proteins interact with each other. Among these, BDBBX8 interacts with a variety of BDBBX proteins, such as BDBBX10, BDBBX11, BDBBX15, and BDBBX21. Meanwhile, BDBBX10, BDBBX11, and BDBBX21 interact with various other proteins, including the SIGE protein, CCA1 protein, AT5G64170 protein, and COP1 protein. These findings suggest that BDBBX proteins play an important role in the BBX protein family of *B. pervariabilis* × *D. grandis*.

Adverse environmental factors such as salt, drought, and cold can severely impact plant growth and development [44]. Previous studies have demonstrated that *BBX* genes have an important role in response to abiotic stress. For example, *AtBBX24* plays a significant role in signaling induced by salt adversity and can enhance the growth of Arabidopsis root systems under high-salinity conditions [45,46]. Similarly, a substantial proportion of *BBX* genes in *Malus domestica* are up-regulated in their expression under osmotic stress, high amounts of salt, low temperatures, and abscisic acid treatment [47]. *CmBBX24* has a dual role in *Dendranthema morifolium*, delaying blooming and enhancing drought or cold tolerance in the plant [48]. The expression of *OsBBX25* in *Oryza sativa* is induced by salt, drought, and ABA, and the overexpression of this gene in *Arabidopsis* enhances the transgenic plant’s resistance to salt and water deficit stresses [23].

Building on these findings, we investigated the expression of *BDBBX* genes in *B. pervariabilis* × *D. grandis* under various abiotic stress conditions. Our experimental results revealed that under water deficit stress conditions, the expression levels of eight genes (*BBX4*, *BBX10*, *BBX11*, *BBX14*, *BBX15*, *BBX16*, *BBX17*, and *BBX21*) were higher than those in the control group for different varieties of hybrid bamboos. These findings suggest an up-regulation trend of gene expression and may indicate that these genes play a vital role in water deficit tolerance in hybrid bamboo. Nine genes, namely *BBX2*, *BBX2*, *BBX3*, *BBX7*, *BBX9*, *BBX10*, *BBX12*, *BBX15*, *BBX16*, and *BBX21*, were found to be expressed at higher levels than the control group under salt stress conditions. These findings suggest that these genes may help in improving the salt tolerance of hybrid bamboo. Among these genes, *BBX10*, *BBX15*, *BBX16*, and *BBX21* exhibit both salt and water deficit tolerance and are important references for the molecular breeding of hybrid bamboo in response to adverse stress conditions. For low-temperature stress, a total of 12 genes (*BBX1*, *BBX2*, *BBX4*, *BBX7*, *BBX10*, *BBX12*, *BBX14*, *BBX15*, *BBX17*, *BBX18*, *BBX19*, and *BBX21*) were expressed at higher levels than the control group, indicating their possible role as cold resistance genes in hybrid bamboo.

*OsBBX1* has been experimentally shown to be responsive to abiotic stresses such as salt and cold stress and exhibits high expression levels under both these conditions [35]. *BDBBX2*, a direct homolog of *OsBBX1* with 100% homology, was also found to be highly expressed under salt and cold stress conditions in three hybrid bamboo varieties. Additionally, *OsBBX24*, which is up-regulated under abiotic stresses such as drought, cold, and salt stress [35], exhibits high homology with *BDBBX4* and *BDBBX21* with 100% and 81.01% similarities in amino acid sequence alignment, respectively. The two *BDBBX* genes’ expression levels were up-regulated under various abiotic stress conditions.

BBX proteins are vital to both biotic and abiotic stress responses in plant growth. In *Oryza sativa*, the knockdown of *OsCOL9* increases the susceptibility of transgenic lines to rice blast, while the overexpression of the gene enhances the resistance of plants to rice blast [23]. To assess the expression of the *BDBBX* gene in hybrid bamboo under biological stress, twelve genes (*BBX1*, *BBX2*, *BBX4*, *BBX7*, *BBX10*, *BBX12*, *BBX14*, *BBX15*, *BBX17*, *BBX18*, *BBX19*, and *BBX21*) were found to exhibit high expression levels in hybrid bamboo infested with *A. phaeospermum*. It is suggested that these *BDBBX* genes may be associated with disease resistance in *B. pervariabilis* × *D. grandis*. Previous studies have indicated that *B. pervariabilis* × *D. grandis* bamboo exhibits less resistance to cold and may experience irregular fluctuations in POD activity, soluble protein content, soluble sugar content, and proline content under low-temperature chilling, which increases the likelihood of disease infestation [3]. Based on the results of low-temperature stress, it was observed that 12 *BDBBX* genes associated with disease resistance were also cold resistant.

A thorough analysis of all the results led to the identification of three genes, namely *BBX10*, *BBX15*, and *BBX21*, which exhibited consistently up-regulated expression levels under various biotic and abiotic stress conditions in hybrid bamboo. These findings suggest that these three genes may play a major role in the response of hybrid bamboo to biotic and abiotic stresses, and further studies on their specific molecular regulatory mechanisms could help improve the biotic and abiotic adversity resistance of hybrid bamboo.

## 4. Materials and Methods

### 4.1. Test Materials

Hybrid bamboo varieties: The *B. pervariabilis* × *D. grandis* bamboo, which was 1 year old and 40–50 cm tall with a ground diameter of 1–1.5 cm, was supplied from a hybrid bamboo cultivation area in Tianquan County, Sichuan Province, China (103°01′N, 29°54′E; altitude, 515.9 m; annual temperature, 16 °C; annual precipitation, 800–1300 mm). Three varieties were chosen for analysis: #3 (moderately resistant), #6 (highly resistant), and #8 (susceptible) [1]. These varieties were provided by the hybrid bamboo cultivation area in Tianquan County, Sichuan Province. An *A. phaeospermum* strain, identified by the registration number OK626768, was also used. This strain was originally isolated from diseased *A. phaeospermum* and was provided by the Sichuan Key Laboratory of Forest Conservation.

To conduct this study, *BBX* nucleic acid sequences of *B. pervariabilis* × *D. grandis* were obtained from a transcriptome database. All interaction transcriptome data for the bamboo samples are available in the NCBI Sequence Read Archive (SRA) under accession numbers SRR14685222, SRR14685221, SRR14685220, SRR14685219, SRR14685218, SRR14685217, SRR14685216, SRR14685215, SRR14685214, SRR14685213, SRR14685212, and SRR14685211. The original data can be accessed in the NCBI Serial Read Archive (SRA) under Biological Project Approval number SAMN19312317 (https://www.ncbi.nlm.nih.gov/biosample/SAMN19312317/ accessed on 1 September 2022). The BBX protein sequence for the plant was downloaded from the GenBank database.

### 4.2. BBX Gene Family Identification and Analysis

To identify BBX protein sequences in the *B. pervariabilis* × *D. grandis* transcriptome database, all plant BBX protein sequences were used as probe sequences. The local BLAST (2.13.0+) software (https://ftp.ncbi.nlm.nih.gov/blast/executables/blast+/LATEST/, accessed on 12 October 2022) was used to search for sequences with similar nucleic acid/protein sequences in the target genome, with a criterion of identity ≥ 50%.

After identifying all possible *BBX* sequences, redundant sequences were removed, and the predicted conserved structural domains of BBX were identified using SMART (v8), Pfam (27.0) online software, and the CD-Search function in Blast. Sequences that did not contain conserved structural domains of the *BBX* gene family were excluded from the analysis.

### 4.3. Analysis of the Basic Physicochemical Properties and Conserved Motifs of the BBX Gene Family

The physicochemical properties of the BDBBX protein sequence were predicted using the ProtParam online analysis tool (http://web.expasy.org/protparam/, accessed on 25 October 2022). The Plant-mPLoc Server (http://www.csbio.sjtu.edu.cn/bioinf/plant-multi/#, accessed on 26 October 2022) was used to predict the subcellular localization of the BDBBX protein. SignalP 4.1 Server (http://www.cbs.dtu.dk/services/SignalP/, accessed on 26 October 2022) was used to predict signal peptides and further identify the subcellular localization of the BDBBX protein.

To predict conserved motifs in the *BDBBX* gene family, the online software MEME4.12.0 (http://meme.nbcr.net/meme/, accessed on 25 October 2022) was used with the maximum number of motifs set to 10 and a motif length range of 6~100 aa, and other parameters were set as default. The conserved motifs were then functionally annotated using the online software SMART (v8) (http://smart.embl-heidelberg.de/, accessed on 15 October 2022).

### 4.4. Phylogenetic Relationship Analysis of the BBX Gene Family

The identified BDBBX protein sequences were compared with reported *BBX* sequences from model plants *Arabidopsis thaliana* and *Oryza sativa* using Clustal X 1.83 software. The results were then used to construct a phylogenetic tree using MEGA 6.0 software. The neighbor-joining (NJ) method was utilized to construct the tree with the P-distance model. The pairwise deletion (gap) option was selected, and the Bootstrap method was set to 1000. The remaining parameters were set as default values.

### 4.5. Structural Information Analysis of the BBX Gene Family

To predict the secondary structure of candidate genes, the online analysis tool SOPMA (https://npsa-prabi.ibcp.fr/cgi-bin/npsa_automat.pl?page=npsa_sopma.html, accessed on 11 November 2022) was used with the structural information of the BDBBX protein. The results of the analysis were presented visually.

For protein tertiary structure prediction analysis, the online analysis tool SWISS-MODEL (https://swissmodel.ex[asy.org/interactive, accessed on 12 November 2022) was used.

### 4.6. GO Analysis of the BBX Gene Family

Functional annotation of the identified BBX proteins was performed and visualized using Blast2GO (https://www.blast2go.com/, accessed on 16 November 2022). Proteins were categorized based on three categories under GO classification, namely biological processes, molecular functions, and cellular components.

### 4.7. Potential Phosphorylation Sites and Glycosylation Analysis of BBX Gene Family

The prediction of potential phosphorylation sites in the *BDBBX* gene family was performed using the NetNGlyc-1.0 online analysis tool (https://services.healthtech.dtu.dk/services/NetNGlyc-1.0/, accessed on 21 November 2022). Analysis of glycosylation sites in the *BDBBX* gene family was performed using the NetPhos 3.1 online analysis software (https://services.healthtech.dtu.dk/services/NetPhos-3.1/, accessed on 21 November 2022).

### 4.8. BDBBX Protein Interaction Network Prediction and Functional Annotation

The hypothetical BDBBX protein sequence for *B. pervariabilis* × *D. grandis* was submitted to the online STRING version 10 (http://string-db.org, accessed on 23 November 2022) and compared to the plant species *Arabidopsis thaliana*. Genes with the highest scores were used to create a hierarchical network of proteins.

### 4.9. Treatment of Biotic and Abiotic Stress Conditions

To characterize the transcriptional profile of the *BBX* gene, *B. pervariabilis* × *D. grandis* bamboo plants were exposed to various biotic and abiotic stresses. Abiotic stressors included water deficit, salt, and cold stress.

For each of the three stress conditions, 12 pots of bamboo plants (40–50 cm tall) were used, with four pots provided for each of the three hybrid bamboo varieties. All the bamboo plants were placed in pure water in a constant temperature incubator at 26 °C, 16 h of light, and 70% humidity for three days. For the water deficit treatment, bamboo plants were transferred to a solution of 20% PEG-6000 and incubated under the same conditions as above. For the salt treatment, bamboo plants were transferred to a 200 mM NaCl solution and incubated under the same conditions as above [35]. For the cold stress treatment, bamboo plants were transferred to a cooler (SANYO) at 10 °C and incubated under the same conditions as above [49]. After 0, 6, 12, and 24 h of each stress treatment, three copies of the bamboo shoot tips weighing 50–100 mg each were taken from each pot and stored frozen at −80 °C for subsequent experiments.

To assess biotic stress response, *B. pervariabilis* × *D. grandis* bamboo plants were subjected to infestation with the fungus *A. phaeospermum*. *A. phaeospermum* was activated on potato dextrose agar plates and incubated at 25 °C for 7 days [2]. The spores were then diluted using sterile saline to prepare a suspension of *A. phaeospermum* spores (10^6^ spores/mL). For the experiment, 45 bamboo plants of uniform growth (40–50 cm in height) were selected, and 15 pots of each of the three hybrid bamboo varieties were placed in a constant temperature incubator under the same conditions for cultivation. The bamboo culms were infested through syringe injection, with 10 mL of spore suspension injected into *B. pervariabilis* × *D. grandis* branches. Inoculations were made at the culm of the branch, and the spore suspension was inoculated twice daily (9 a.m. and 9 p.m.) and treated with gauze for moisture. After 0, 3, 5, 7, and 10 days of pathogen infestation treatments, three copies of bamboo shoot tips weighing 50–100 mg each were taken from each pot and stored frozen at −80 °C for subsequent experiments. Meanwhile, the incidence rate and disease index of hybrid bamboo at different periods of pathogen infestation were counted. Spearman’s correlation coefficient of the website (http://www.cloudtutu.com/#/login, accessed on 25 August 2023) was used to further correlate the gene expression with incidence rate and the disease index after pathogen infestation. Refer to Appendix A for the grading standard of the disease index. The disease index and incidence rate are calculated according to the following formula.
Disease index = [Σ(numerical value of each disease rating × number of plants in each disease rating)/(total plants × most serious disease rating)] × 100
Incidence rate (%) = Number of diseased plants/Total number of plants × 100

### 4.10. Quantitative PCR Analysis

After treating the collected bamboo shoot tips under experimental conditions, they were rapidly ground in liquid nitrogen, and RNA was extracted [6]. The purity and concentration of the extracted RNA were assessed using a NanoDrop 2000 ultraviolet spectrophotometer. Gene stability was tested via 1% agarose gel electrophoresis, and the RNA was stored in a refrigerator at −80 °C. The cDNA was reverse transcribed in accordance with the instructions provided in the reverse transcription kit, and the resulting cDNA was stored in a −20 °C refrigerator for quantitative real-time PCR (qRT-PCR) analysis. Using Primer Premier version 5, 21 gene primers (Appendix A) for the hybrid bamboo *BDBBX* gene were synthesized based on the information obtained from the coding sequences. To verify the specificity of each primer pair, they were generated from a conserved domain of a gene, and their specificity was confirmed using the BLAST function of NCBI.

For qRT-PCR analysis, we utilized the SYBR^®^Premix Ex Taq™ (Tli RNaseH Plus) from TaKaRa Biotechnology Ltd. (Dalian, China) in conjunction with an Applied Biosystems 7500 real-time PCR instrument. A reaction mixture of 20 μL was created containing 1.0 μL of cDNA template, 0.4 μM of each primer (F/R), 10 μL of 2xQ3 SYBR qPCR Master Mix, and 8.2 μL of RNase-free water, as per the manufacturer’s guidelines. The PCR conditions were implemented according to the specified protocol for SYBR^®^Premix Ex Taq™. To minimize errors, three replicates of each experiment were conducted. Relative transcript abundance values were computed through the 2^−∆∆Ct^ method [50].

### 4.11. Comparison of Amino Acid Sequences of BBX Genes from Different Species

For the purpose of aligning amino acid sequences across multiple species, BDBBX proteins that showed a response to all stressors were chosen. We used BLAST-P (https://blast.ncbi.nlm.nih.gov, accessed on 4 December 2022) to compare the responsive *BDBBX* gene to other Gramineae sequences. Subsequently, the *BBX* gene sequences of the top three Gramineae crops with the highest scores for each gene pair were downloaded via NCBI and classified into the same group as the corresponding *BDBBX* genes. The downloaded sequences were translated using the online software ExPASy (3.0) translation tool (https://web.expasy.org/translate, accessed on 5 December 2022). Using the CLUSTALW multiple sequence alignment (https://www.genome.jp/tools-bin/clustalw, accessed on 5 December 2022), we compared the three sets of *BBX* amino acid sequences, and the results were analyzed using Jalview (9.0.5) software (https://www.jalview.org/, accessed on 6 December 2022).

## 5. Conclusions

This study aimed to identify and analyze 21 members of the *BBX* gene family using sequencing data from the transcriptome of *B. pervariabilis* × *D. grandis*. Our results revealed a high degree of conservation in the sequence structure of the *BBX* gene family in *B. pervariabilis* × *D. grandis*, and they were classified into four subfamilies based on conserved structural domains. The bioinformatic analysis of their gene structure, conserved motifs, phylogeny, and predicted protein interactions provided us with a comprehensive understanding of the structural relationships among the family members. The analysis of biotic and abiotic stress on the *BDBBX* gene predicted three key functional *BBX* genes (*BDBBX10*, *BDBBX15*, and *BDBBX21*), which will serve as a theoretical basis for an in-depth analysis of the molecular regulatory mechanism of resistance to disease in *B. pervariabilis* × *D. grandis*.

## Figures and Tables

**Figure 1 ijms-24-13465-f001:**
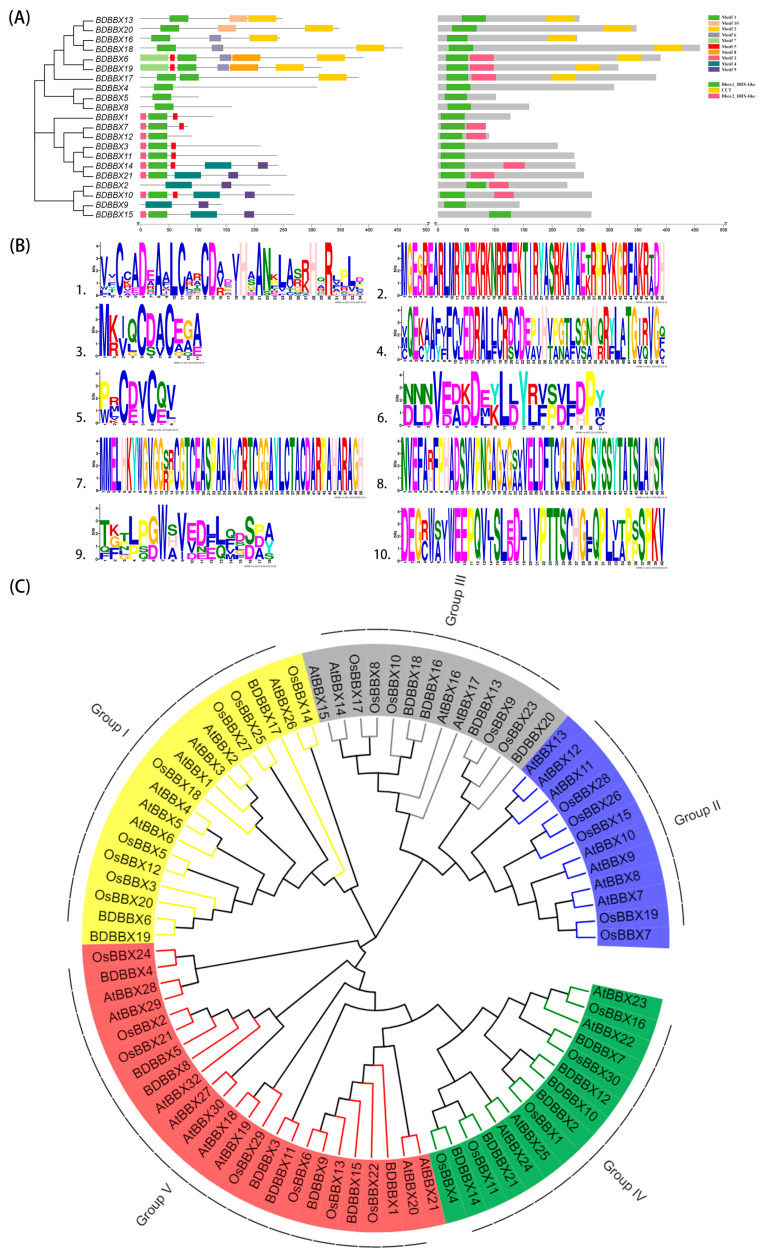
The phylogenetic tree, protein domain, conserved motifs and their sequence information within the BDBBX protein family, and the phylogenetic tree of relationships between members of the BBX protein family in *Arabidopsis thaliana*, *Oryza sativa*, and *Bambusa pervariabilis* × *Dendrocalamopsis grandis*. (**A**) Phylogenetic tree, protein structural domains, conserved motifs of BDBBX protein; and (**B**) Sequence information of conserved motifs of BDBBX protein. Sequence information includes each conserved sequence’s length, the conservation degree, and the amino acid’s sharing or substitution. Each motif has a corresponding axis, with the horizontal axis representing the corresponding position of each amino acid on the pattern and the vertical axis representing the sections. (**C**) Phylogenetic tree of the neighbor-joining method for the BBX proteins of *Bambusa pervariabilis* × *Dendrocalamopsis grandis*, *Arabidopsis thaliana*, and *Oryza sativa*. Classes of different colors represented different subfamilies of *Bambusa pervariabilis* × *Dendrocalamopsis grandis*.

**Figure 2 ijms-24-13465-f002:**
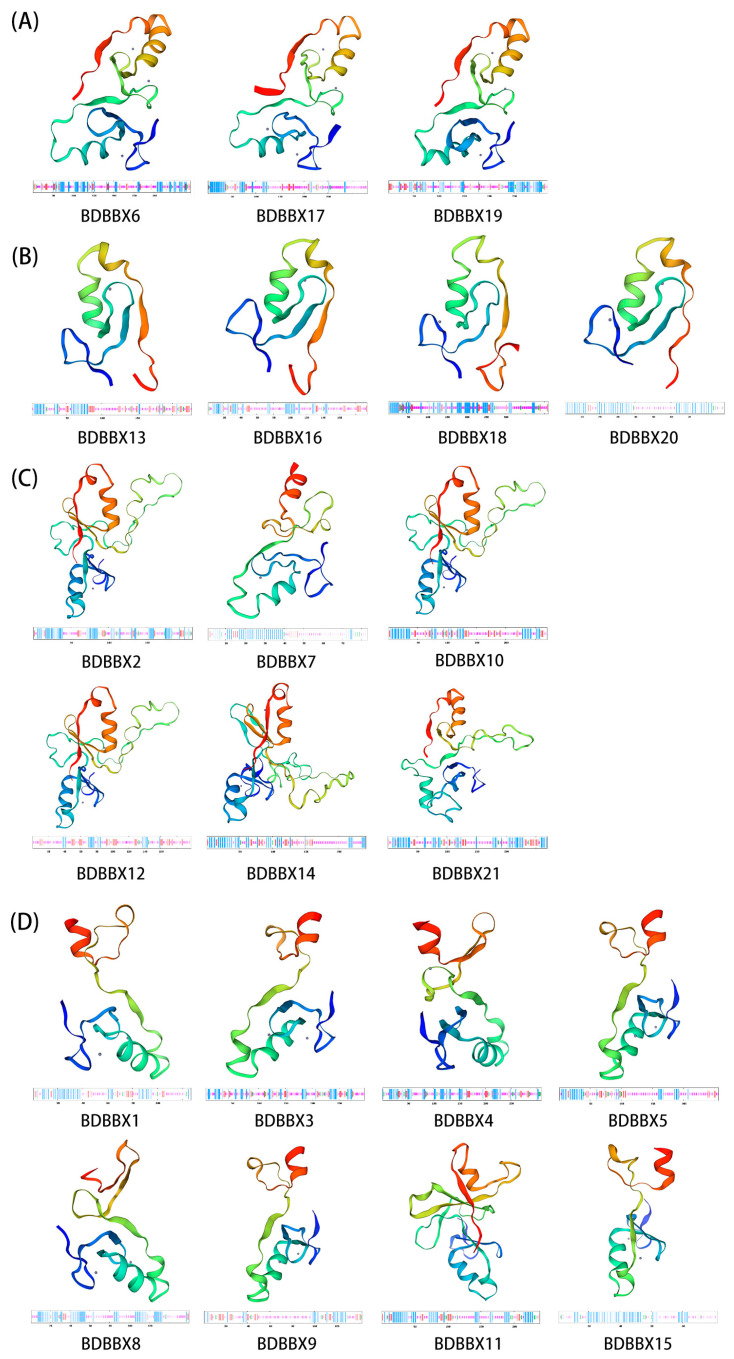
Secondary and tertiary structures of BBX proteins. (**A**) is Group I, (**B**) is Group III, (**C**) is Group IV, and (**D**) is Group V. The sequence of the protein is segmented from n to c, and the color transitions from cool (blue) to warm (red), with different colors representing different helices and folding regions.

**Figure 3 ijms-24-13465-f003:**
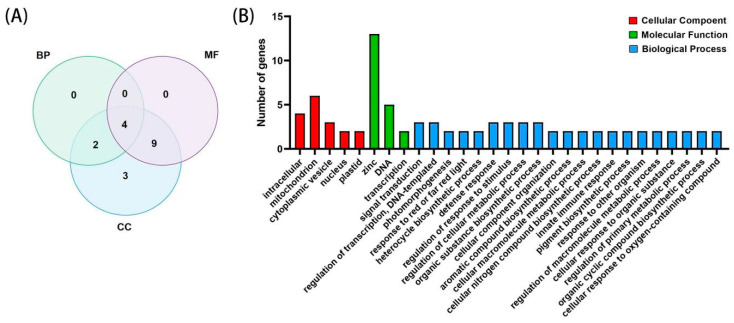
GO annotation of the *BDBBX* gene. (**A**) shows the Venn diagram of the distribution of *BDBBX* genes in the biological process (BP), molecular function (MF), and cellular component (CC) functional categories. (**B**) shows the subcategories of the GO classification of the *BDBBX* gene (Level 2), where the *x*-axis represents the number of genes, and the *y*-axis represents the name of the GO subcategory function.

**Figure 4 ijms-24-13465-f004:**
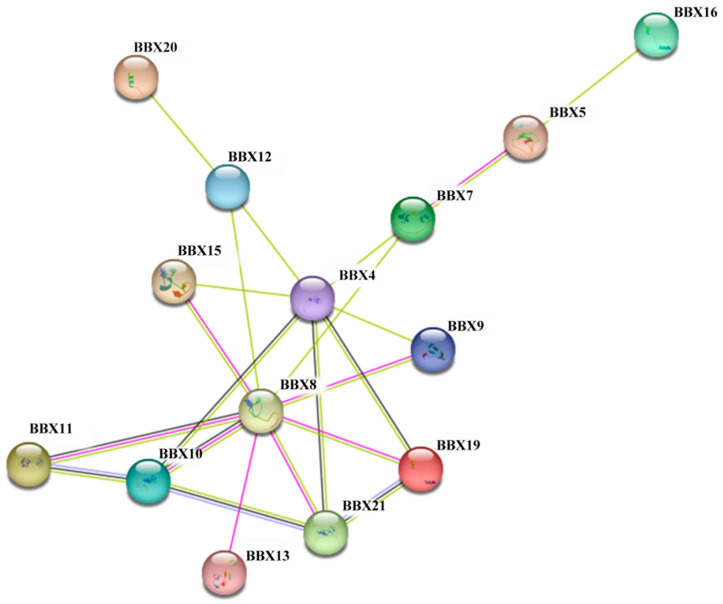
BDBBX gene family protein–protein interaction network. The connecting lines of protein interactions have different meanings due to their different colors. Yellow represents text mining, black represents co-expression, purple represents protein physiology, and pink represents experimentally determined.

**Figure 5 ijms-24-13465-f005:**
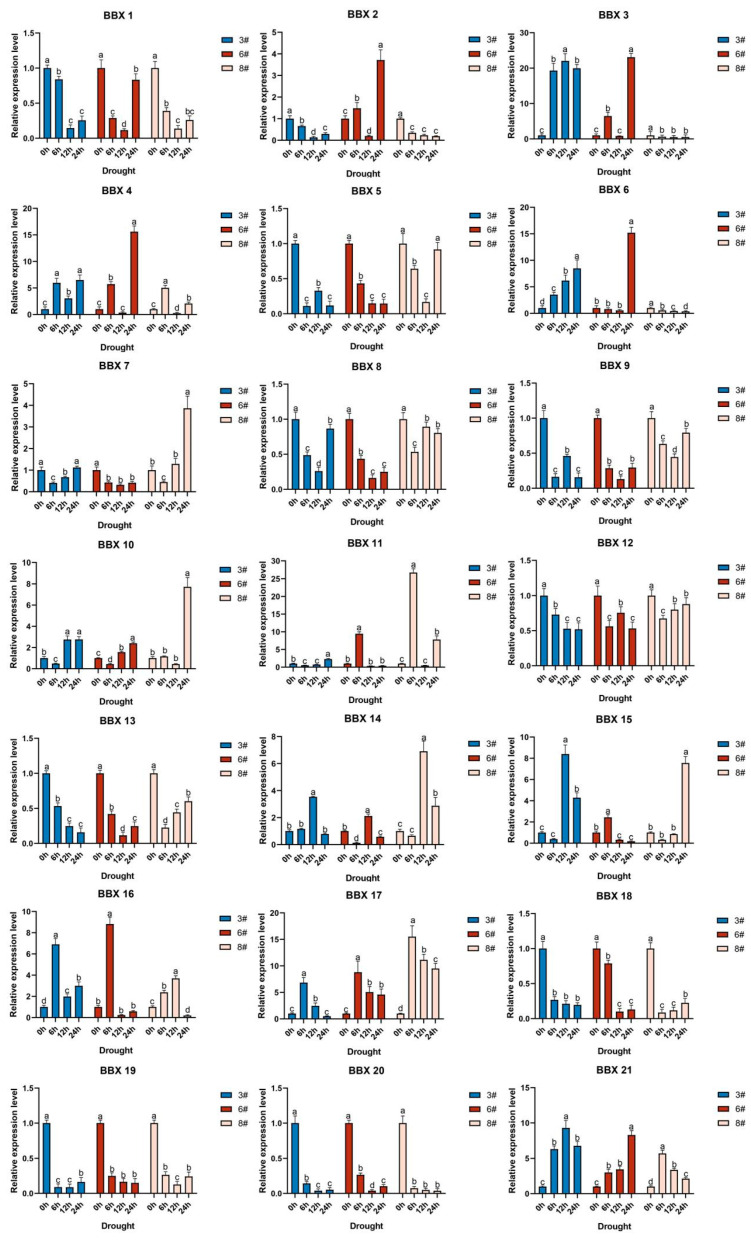
Changes in the expression of 21 *BDBBX* genes over time in three varieties of hybrid bamboo under water deficit stress. Histogram showing the changes in the expression of 21 *BDBBX* genes over time in three varieties of hybrid bamboo under water deficit stress. The hybrid bamboo was stressed using a 20% concentration of PEG-6000 solution. Leaves of the plants were taken after 0 h, 6 h, 12 h, and 24 h, and the expression of the hybrid bamboo *BBX* genes was detected by qPCR testing. The relative expression of the target genes was calculated using the 2^−∆∆Ct^ method. The horizontal coordinates indicate each sampling time point of the experiment, and the vertical coordinates indicate the relative expression levels. Different lowercase letters indicate the significance of differences between treatments at the 0.005 level (*p* < 0.05).

**Figure 6 ijms-24-13465-f006:**
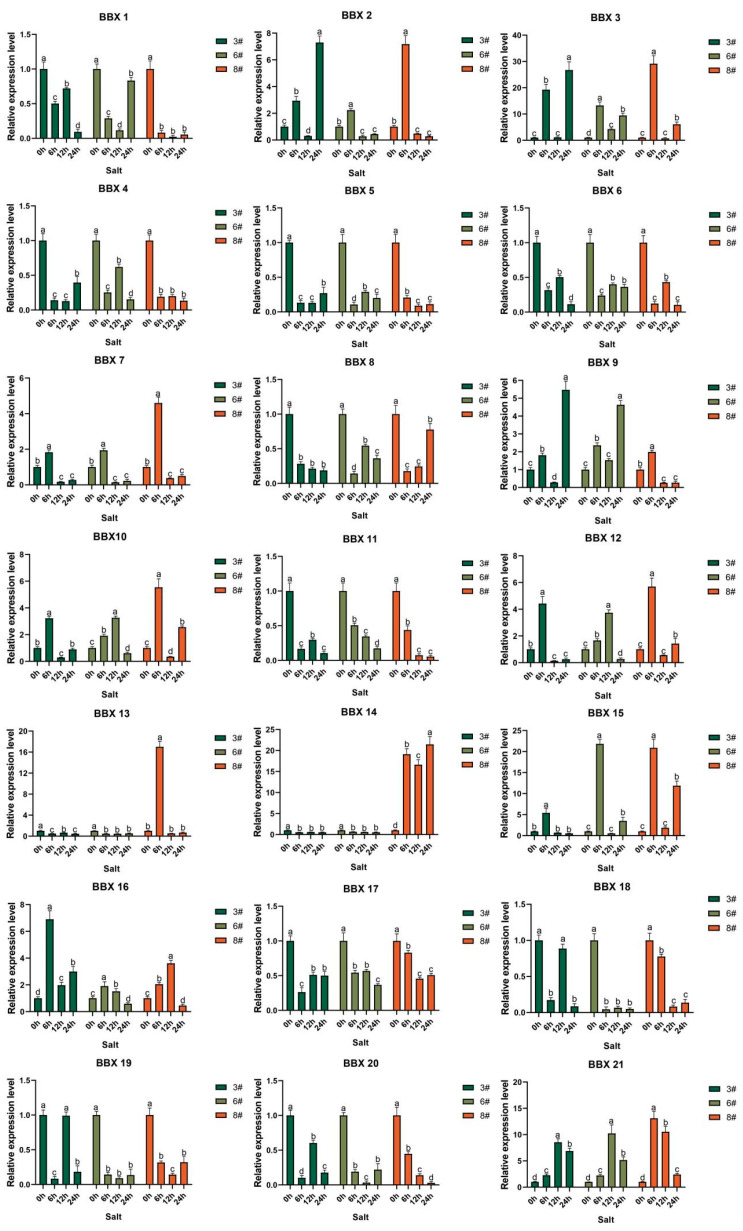
Changes in the expression of 21 *BDBBX* genes over time in three varieties of hybrid bamboo under salt stress. Histogram showing the changes in the expression of 21 *BDBBX* genes over time in three varieties of hybrid bamboo under salt stress. The hybrid bamboo was stressed using 200 mM NaCl solution. Leaves of the plants were taken after 0 h, 6 h, 12 h, and 24 h, and the expression of the hybrid bamboo *BBX* genes was detected by qPCR testing. The relative expression of the target genes was calculated using the 2^−∆∆Ct^ method. The horizontal coordinates indicate each sampling time point of the experiment, and the vertical coordinates indicate the relative expression levels. Different lowercase letters indicate the significance of differences between treatments at the 0.005 level (*p* < 0.05).

**Figure 7 ijms-24-13465-f007:**
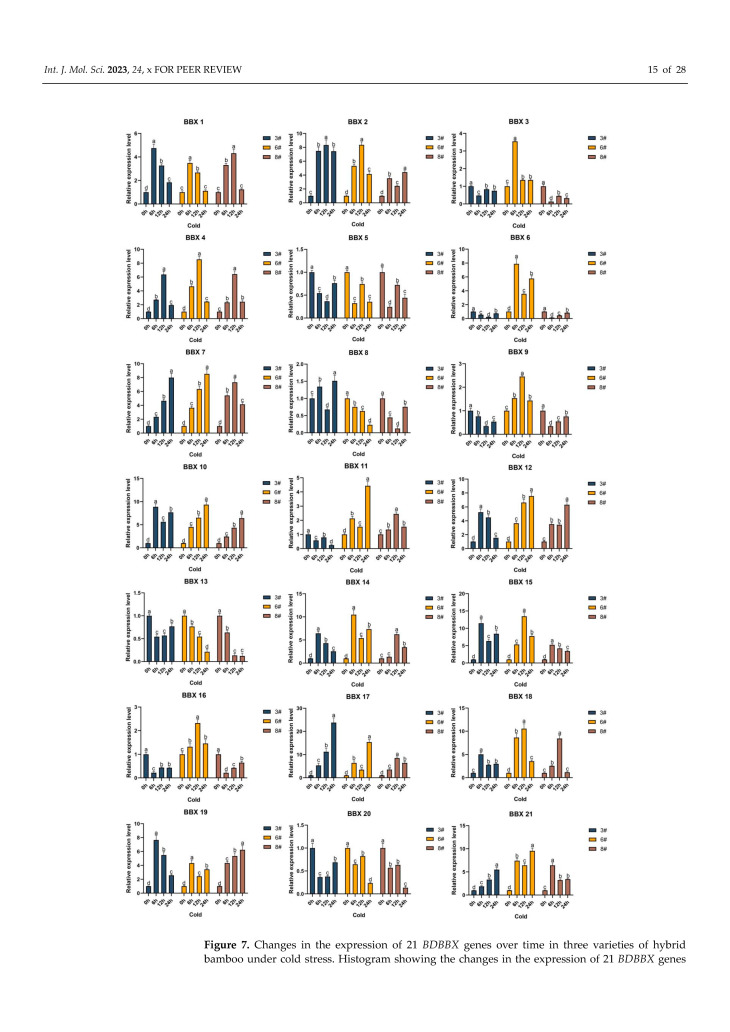
Changes in the expression of 21 *BDBBX* genes over time in three varieties of hybrid bamboo under cold stress. Histogram showing the changes in the expression of 21 *BDBBX* genes over time in three varieties of hybrid bamboo under cold stress. Transfer the hybrid bamboo to a cooler (SANYO) at 10 °C for stress. Leaves of the plants were taken after 0 h, 6 h, 12 h, and 24 h, and the expression of the hybrid bamboo *BBX* genes was detected by qPCR testing. The relative expression of the target genes was calculated using the 2^−∆∆Ct^ method. The horizontal coordinates indicate each sampling time point of the experiment, and the vertical coordinates indicate the relative expression levels. Different lowercase letters indicate the significance of differences between treatments at the 0.005 level (*p* < 0.05).

**Figure 8 ijms-24-13465-f008:**
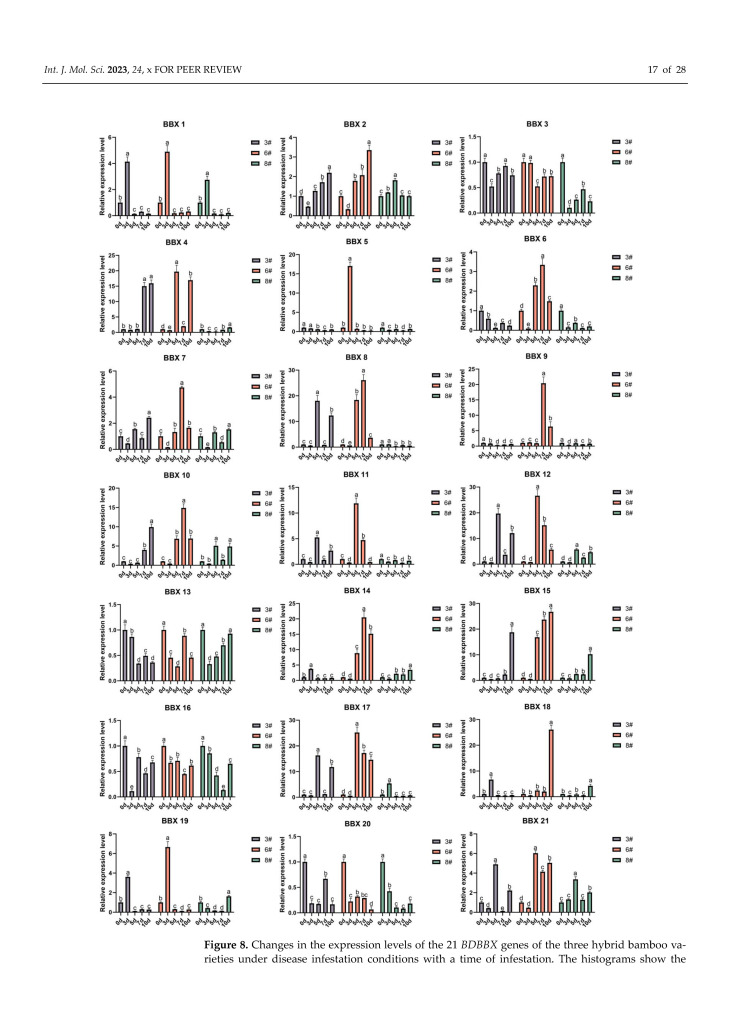
Changes in the expression levels of the 21 *BDBBX* genes of the three hybrid bamboo varieties under disease infestation conditions with a time of infestation. The histograms show the changes in the expression of the 21 *BDBBX* genes over time in the different hybrid bamboo varieties under disease infestation. Hybrid bamboos were infested by injection of the spore suspension of the pathogen, and leaves of the plants were taken after 0 d, 3 d, 5 d, 7 d, and 10 d. The expression of the hybrid bamboo *BBX* genes was detected by qPCR testing. The relative expression of the target genes was calculated using the 2^−∆∆Ct^ method. The horizontal coordinates indicate each sampling time point of the experiment, and the vertical coordinates indicate the relative expression levels. Different lowercase letters indicate the significance of differences between treatments at the 0.005 level (*p* < 0.05).

**Figure 9 ijms-24-13465-f009:**
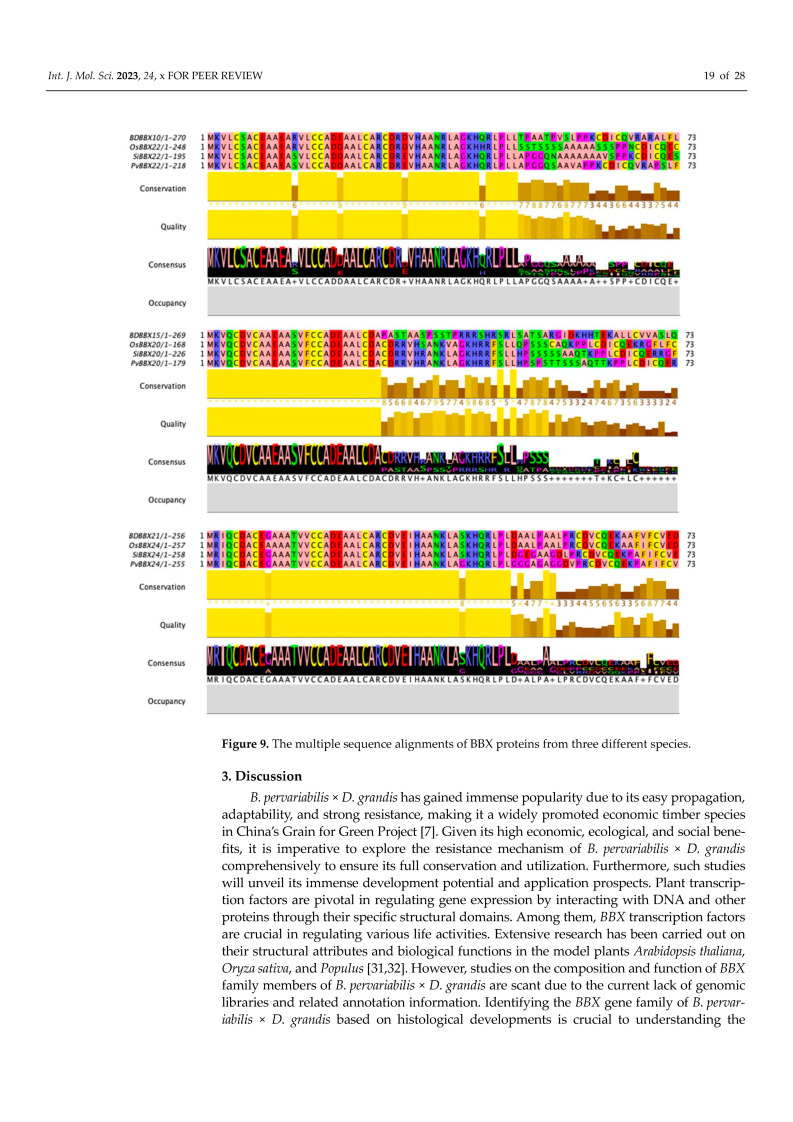
The multiple sequence alignments of BBX proteins from three different species.

**Table 1 ijms-24-13465-t001:** BBX gene family information of *Bambusa pervariabilis* × *Dendrocalamopsis grandis*.

Transcriptome ID	Gene	Protein Length (AA)	Molecular Weight (Da)	Theoretical pI	Instability Index	Aliphatic Index	Grand Average of Hydropathicity (GRAVY)	Signal Peptide	Subcellular Localization
PH01000042G1610	*BDBBX1*	127	13,945.96	10.26	70.44	60.08	−0.521	NO	Nucleus
PH01000149G0140	*BDBBX2*	227	24,305.66	4.82	59.77	80.84	−0.013	NO	Nucleus
PH01000192G1360	*BDBBX3*	210	22,874.85	11.57	63.94	90.29	−0.337	NO	Nucleus
PH01000303G0990	*BDBBX4*	309	32,565.29	6.41	55.53	58.25	−0.518	NO	Chloroplast.Nucleus
PH01000450G1030	*BDBBX5*	102	10,539.61	8.66	74.10	51.08	−0.52	NO	Chloroplast. Cytoplasm
PH01000481G0620	*BDBBX6*	383	41,494.94	7.59	41.16	67.44	−0.295	NO	Cell membrane.Nucleus
PH01000616G0330	*BDBBX7*	83	8412.52	6.25	56.72	71.93	−0.014	NO	Nucleus
PH01000917G0300	*BDBBX8*	160	16,681.41	10.50	73.87	62.62	−0.358	NO	Nucleus
PH01001451G0380	*BDBBX9*	143	14,723.24	4.69	61.26	67.69	−0.113	NO	Nucleus
PH01001725G0310	*BDBBX10*	270	28,776.28	7.48	50.37	73.89	−0.087	NO	Nucleus
PH01001870G0430	*BDBBX11*	239	26,213.84	11.48	63.61	86.65	−0.431	NO	Nucleus
PH01002961G0180	*BDBBX12*	90	10,098.74	9.23	56.06	58.78	−0.371	NO	Nucleus
PH01003421G0140	*BDBBX13*	198	21,263.84	5.30	52.63	64.70	−0.366	NO	Nucleus
Phyllostachys_edulis_newGene_30773	*BDBBX14*	241	26,221.19	7.86	41.89	81.08	−0.126	NO	Nucleus
Phyllostachys_edulis_newGene_38997	*BDBBX15*	269	29,499.68	6.45	60.16	82.75	−0.099	NO	Nucleus
PH01000037G0060	*BDBBX16*	194	20,980.48	9.49	79.35	55.46	−0.699	NO	Nucleus
PH01000780G0510	*BDBBX17*	332	36,625.69	11.90	93.79	41.08	−0.932	NO	Chloroplas.
PH01002146G0170	*BDBBX18*	409	44,318.82	6.43	63.00	58.58	−0.593	NO	Nucleus
PH01002727G0080	*BDBBX19*	316	34,063.68	6.08	37.30	74.56	−0.199	NO	Nucleus
Phyllostachys_edulis_newGene_62371	*BDBBX20*	298	32,384.84	6.81	59.12	74.03	−0.345	NO	Nucleus
PH01003160G0610	*BDBBX21*	256	27,100.62	4.96	50.25	76.33	−0.155	NO	Nucleus

**Table 2 ijms-24-13465-t002:** Conserved motifs of BDBBX proteins and their functional annotations.

Motif	Motif Length (AA)	Motif Sequence	Function Annotation
1	34	VVCCADEAALCARCDADVHAANKLASRHQRLPLD	zf-B_box
2	50	MGEGREARLMRYREKRKNRRFEKTIRYASRKAYAEKRPRIKGRFAKRADH	CCT
3	11	MKIQCDACEGA	UnKnown
4	47	VQEKAAFIFCVEDRALLCRDCDEPIHVPGTLSGNHQRYLATGIRVGF	zf-B_box
5	8	PRCDVCQV	UnKnown
6	21	DLDVEDDDEKLDYRFPDFDPY	UnKnown
7	50	MMELHKYWGVGGRRCGTCEASPAAVHCRTCGGAYLCTACDARPAHARAGH	zf-B_box
8	50	NVEFARFPHADSVVPNGAGVGAVVELDFTCGLGAKPSYSSYTATSLAHSV	UnKnown
9	18	TGTLPGWAVEDLLFDSPA	UnKnown
10	40	DEGCWAIWEEPQVJSLEDJIVPTTSCHGFQPLLAPPSPKV	UnKnown

**Table 3 ijms-24-13465-t003:** Predicted information on protein kinase phosphorylation sites and glycosylation sites.

Gene ID	Phosphorylation Sites			Glycosylation Sites			
	Serine	Threonine	Tyrosine	Position	Potential	Jury Agreement	N-Glyc Result
BDBBX1	9	5	/	None			
BDBBX2	13	7	1	105 NASA	0.5176	(4/9)	+
				216 NGTS	0.669	(8/9)	+
BDBBX3	2	/	1	None			
BDBBX4	27	6	2	299 NCSN	0.5557	(5/9)	+
BDBBX5	11	1	1	None			
BDBBX6	17	9	6	None			
BDBBX7	4	/	2	None			
BDBBX8	12	3	3	None			
BDBBX9	16	5	1	70 NSSS	0.57	(8/9)	+
BDBBX10	20	7	1	209 NRTR	0.5787	(6/9)	+
BDBBX11	5	2	3	None			
BDBBX12	5	1	1	None			
BDBBX13	6	8	/	None			
BDBBX14	14	9	2	234 NLTL	0.6876	(9/9)	++
BDBBX15	25	12	3	152 NSSS	0.611	(8/9)	+
BDBBX16	21	5	3	None			
BDBBX17	30	28	/	200 NSSA	0.5469	(6/9)	+
BDBBX18	29	8	1	None			
BDBBX19	10	5	6	None			
BDBBX20	18	15	1	179 NSTL	0.6232	(8/9)	+
				227 NSTE	0.5468	(6/9)	+
BDBBX21	10	6	3	None			

## Data Availability

All interaction transcriptome data for the bamboo samples are available in the NCBI Sequence Read Archive (SRA) under accession numbers SRR14685222, SRR14685221, SRR14685220, SRR14685219, SRR14685218, SRR14685217, SRR14685216, SRR14685215, SRR14685214, SRR14685213, SRR14685212, and SRR14685211. The original data can be accessed in the NCBI Serial Read Archive (SRA) under Biological Project Approval number SAMN19312317 (https://www.ncbi.nlm.nih.gov/biosample/SAMN19312317/ accessed on 1 September 2022).

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
