# Peer review of "Identification and Characterization of the BBX Gene Family in Bambusa pervariabilis × Dendrocalamopsis grandis and Their Potential Role under Adverse Environmental Stresses"

_ijms, 2023, doi:10.3390/ijms241713465_

Round 1
Reviewer 1 Report
This paper deals with the investigation of the BBX family genes in bamboo bioinformatic analysis and qPCR experiments. The authors revealed the expression profiles of these genes under biotic and abiotic stress conditions. Interestingly, a few of these genes showed the different expression patterns among different cultivars. The results presented in this paper will be useful for the development of elite lines that show the enhanced both biotic and abiotic stress tolerance.
I have only one minor comment. Line 89, "abscisic acid stress" seems to be inappropriate.
Author Response
Modification instructions
Dear doctor,
Thank you for your comments concerning our manuscript entitled “Identification and characterization of the BBX gene family in Bambusa pervariabilis × Dendrocalamopsis grandis and their potential role under adverse environmental stresses”.
Those comments are all valuable and very helpful for revising and improving our paper, as well as the important guiding significance to our research. Revised portions are marked in the paper. The main corrections in the paper are as follows:
To question : Line 89, "abscisic acid stress" seems to be inappropriate.
Answer: We appreciate your important comment. According to the suggestion of the reviewer, we have revised it to lines 98-99 “Oryza sativa in response to drought, high salt, low-temperature stress and act on abscisic acid signal transduction processes [27];”

Reviewer 2 Report
Dear Authors
I read with great interest the manuscript of Yi Liu et al "Identification and characterization of the BBX gene family in Bambusa pervariabilis × Dendrocalamopsis grandis and their potential role under adverse environmental stresses".
The authors have done extensive work on the identification of B-box (BBX) genes of Bambusa pervariabilis × Dendrocalamopsis grandis, analyzing their structure, conserved motifs, phylogeny, and putative biological functions of individual members of this family of transcription factors. The analyzed B-box genes were found to be evolutionarily conserved, as evidenced by their comparative analysis with similar gene families in other species, primarily Arabidopsis, rice, and poplar. In spite of the fact that this paper failed to obtain fundamentally new scientific knowledge about BBX genes, the authors have thoroughly characterized the mentioned family of genes in bamboo plants, which have important economic, social and ecological significance. This fact makes the work relevant. I have no doubt at all that the manuscript can be published after some refinements and perhaps a little additional information.
Below are my comments, which are primarily related to the expression of BBX genes under the conditions of abiotic and biotic factors.
1. The authors throughout this paper attempt to link changes in the transcript levels of individual BBX genes to plant resistance to drought, salinity, cold, and A. phaeospertmum infection. In doing so, they identify genes that show an increased intensity of expression under stress conditions, and on this basis they suggest that these genes are the ones that are relevant to the enhancement of a particular type of resistance. It seems to me that these data only allow to state changes in transcript levels of a particular gene under any of the stressors used. In order to look for correlations of gene expression with resistance, one must have data on salt, drought, and salinity tolerance of the study site. I believe that this is the main shortcoming of the paper. I do not exclude that the authors have data on the levels of resistance of the used bamboo hybrids to the mentioned damaging factors.
An exception is an attempt to correlate the expression of BBX genes with plant resistance to A. phaeospertmum infection. In this case, the authors used hybrids with different resistance to the biopathogen: highly resistant #6, moderately resistant #3, and susceptible #8. They further write, "Notably, 12 genes, namely BBX1, BBX2, BBX4, BBX7, BBX10, BBX12, BBX14, BBX15, BBX17, BBX18, BBX19, and BBX21, showed higher expression levels across all three varieties compared to the control group, signifying an up-regulation trend. These genes are presumably the most probable candidates for disease resistance in the hybrids" (lines 392-395). It seems to me that if the level of gene expression increases independently of the resistance level of the variety, then such genes may have nothing to do with resistance.
2. I would also like to know how the authors selected the intensity of the acting factors - salinity, cold and water deficit? Also, strictly speaking, the authors did not study the effects of drought per se, but of water deficit. Polyethylene glycol causes osmotic stress in plants, resulting in water deficit, whereas drought is a more complex factor.
Kind regaards
Author Response
Modification instructions
Dear doctor,
Thank you for your comments concerning our manuscript entitled “Identification and characterization of the BBX gene family in Bambusa pervariabilis × Dendrocalamopsis grandis and their potential role under adverse environmental stresses”.
Those comments are all valuable and very helpful for revising and improving our paper, as well as the important guiding significance to our research. Revised portions are marked in the paper. The main corrections in the paper are as follows:
To question 1: The authors throughout this paper attempt to link changes in the transcript levels of individual BBX genes to plant resistance to drought, salinity, cold, and A. phaeospertmum infection. In doing so, they identify genes that show an increased intensity of expression under stress conditions, and on this basis they suggest that these genes are the ones that are relevant to the enhancement of a particular type of resistance. It seems to me that these data only allow to state changes in transcript levels of a particular gene under any of the stressors used. In order to look for correlations of gene expression with resistance, one must have data on salt, drought, and salinity tolerance of the study site. I believe that this is the main shortcoming of the paper. I do not exclude that the authors have data on the levels of resistance of the used bamboo hybrids to the mentioned damaging factors.
An exception is an attempt to correlate the expression of BBX genes with plant resistance to A. phaeospertmum infection. In this case, the authors used hybrids with different resistance to the biopathogen: highly resistant #6, moderately resistant #3, and susceptible #8. They further write, "Notably, 12 genes, namely BBX1, BBX2, BBX4, BBX7, BBX10, BBX12, BBX14, BBX15, BBX17, BBX18, BBX19, and BBX21, showed higher expression levels across all three varieties compared to the control group, signifying an up-regulation trend. These genes are presumably the most probable candidates for disease resistance in the hybrids" (lines 392-395). It seems to me that if the level of gene expression increases independently of the resistance level of the variety, then such genes may have nothing to do with resistance.
Answer 1: We appreciate your important comment.
About the study site, we cultured all the bamboo plants in pure water in the constant temperature incubator, and all environmental factors were under manual control, 26°C, 70% humidity, and all treatments under the same conditions, with the following modifications for the sake of non-ambiguity, lines 739-740: “All the bamboo plants were placed in pure water in a constant temperature incubator at 26°C, 16 hours of light, and 70% humidity for three days. ” Our transcriptome data were obtained by sequencing the hybrid bamboo after treating the spore suspension of Arthrinium phaeospermum, through which we further mined the BBX gene family of functional genes to be investigated and further verified the conjecture of its disease resistance and stress tolerance by qPCR experiments. During the experiment, we ensured that the hybrid bamboo was subjected to a single stressful environmental change by keeping all the irrelevant variables consistent, thus ensuring that the changes in the expression level of the BBX genes were not interfered with by other factors and acted only on a single environmental variable suffered by the plant.
Furthermore, the one-year-old bamboo plants were provided by the hybrid bamboo cultivation area in Tianquan County, Sichuan Province, China. According to the suggestion of the reviewer, we have made additions in lines 650-653, “The B. pervariabilis × D. grandis bamboo, which was 1 year old and measured 40-50cm tall with a ground diameter of 1-1.5cm, was supplied from a hybrid bamboo cultivation area in Tianquan County, Sichuan Province, China (103°01′N, 29°54′E; altitude, 515.9m; annual temperature, 16°C; annual precipitation, 800–1300 mm). ” This site was chosen because the Tianquan hybrid bamboo cultivation area is at an altitude of about 515.9 m above sea level, with an annual temperature of about 16 °C and an annual precipitation in the range of 800-1300 mm, which is very suitable for supporting the growth of Bambusa pervariabilis × Dendrocalamopsis grandis. Bambusa pervariabilis × Dendrocalamopsis grandis is a high-yielding hybrid bamboo strain, which has higher economic and ecological value than other bamboo species. We want to promote the introduction of this species in more areas, especially in Sichuan including the neighbouring areas with saline soil. Based on this goal, we conducted a large number of stress experiments to screen out broad-spectrum resistance genes to provide a basis for subsequent molecular breeding.
Answer 2: We appreciate your important comment.
According to the suggestion of the reviewer, for the second question, with the known resistance of the three hybrid bamboo varieties, we aimed to screen for significantly highly expressed BBX genes in high resistance, and to make the presentation clearer, we separated the significantly highly expressed BBX genes in the three varieties as follows, lines 415- 440
“The results of disease index and incidence rate showed that (Figure S2), among the three hybrid bamboo varieties, the disease index and incidence rate ranged from low to high by the pattern of high resistance 6# to medium resistance 3# to susceptible 8#. The disease index of high resistance 6# was the lowest at 8.33 and the incidence rate was 16.67%. The susceptible 8# had a higher disease index of 27.78 and an incidence of 50.00%. Interestingly, we observed that the proportion of down-regulated genes in the three varieties correlated with their degree of resistance: five (highly resistant #6), seven (moderately resistant #3), and eight (susceptible #8) down-regulated genes. Meanwhile, among the genes with up-regulated gene expression levels, it was found that the high and low gene expression levels coincided with the pattern of high resistance varieties being higher than medium resistance varieties being higher than susceptible varieties. For example, the expression level of BBX10 was 15 times higher than the control level in variety 6, while the expression level of this gene was 10 times and 5 times higher than the control level in varieties 3 and 8, respectively. The expression level of BBX15 was 25 times higher than the control level in variety 6, while the expression level of this gene was 18 times and 9 times higher than the control level in varieties 3 and 8, respectively. These results indicate that the increase in the expression level of the BBX gene is positively correlated with the resistance of the varieties. Notably, 12 genes, namely BBX1, BBX2, BBX4, BBX7, BBX10, BBX12, BBX14, BBX15, BBX17, BBX18, BBX19, and BBX21, showed a trend of up-regulation of the expression level in all the three varieties as compared to the control. These genes are presumably the most probable candidates for disease resistance in the hybrids. In addition, the results of Figure S2 showed that 12 disease resistance candidate genes were negatively correlated with the incidence rate, among which the gene expression levels of BBX10 and BBX21 were significantly negatively correlated with the incidence rate.”
Table S2 Disease grading criteria
Degree |
Grading criteria |
0 level |
no disease observed on the leaf |
1 level |
disease spots accounting for <30% of the total area |
2 level |
disease spots accounting for 30 to 50% of the total area |
3 level |
disease spots accounting for >50% of the total area |
4 level |
withered leaves |
Disease index = [Σ(numerical value of each disease rating × number of plants in each disease rating)/(total plants × most serious disease rating)] × 100
Incidence rate (%) = Number of diseased plants/Total number of plants × 100
Figure S2 (A) Disease index of three B. pervariabilis × D. grandis varieties; (B) Correlation analysis of gene expression with incidence rate (Ir) and disease index (Di). * indicates p value less than 0.05, ** indicates p value less than 0.01, and *** indicates p value less than 0.001.
To question 2: I would also like to know how the authors selected the intensity of the acting factors - salinity, cold and water deficit? Also, strictly speaking, the authors did not study the effects of drought per se, but of water deficit. Polyethylene glycol causes osmotic stress in plants, resulting in water deficit, whereas drought is a more complex factor.
Answer: We appreciate your important comment. According to the suggestion of the reviewer, we have made the following adjustments,
The experimental methodology was determined through multiple publications, and the solution strengths were referenced in light of the experimental approaches taken to study BBX genes in other species. For example, Wen et al. (2020) used a 20% PEG-6000 solution to study the expression pattern of the PhBBX gene in Petunia hybrida in response to drought stress. Shalmani et al. (2019) also used 20% PEG-6000 solution and 200 mM NaCl solution to stress BBX genes in five grass species (rice, sorghum, stiff brome, Millet, maize).
According to previous studies, it was found that drought experiments are generally performed by using PEG solution to simulate or direct drought treatment. For example, Linghu et al. (2023) in order to investigate the role of Plant U-box (PUB) proteins in the oilseed rape ZS11 cultivar under drought stress, a 20% PEG6000 solution was used for hydroponics to simulate drought stress treatment. Wu et al. (2022) studied the abiotic stress conditions of the WRKY gene in other species and chose to simulate drought in a 15% PEG-6000 solution and cold in a 4 degree Celsius freezer. In order to more accurately control the experimental seedlings experiencing drought stress without being affected by other environmental factors, we decided to perform the experiments in a way that simulates drought conditions by using a 20% solution of PEG-6000.
Given that the simulated drought treatment experiments were described as drought stress by previous authors, we also used drought stress to describe it in our article writing, and based on expert suggestion, we changed the full text from drought stress to water deficit stress.
Wen, SY, Zhang, Y, Deng, Y, Chen, GJ, Yu, YX, Wei, Q. Genomic identification and expression analysis of the BBX transcription factor gene family in Petunia hybrida. Molecular biology reports 2020, 47, 8. https://doi.org/10.1007/s11033-020-05678-y
Shalmani, A, Jing, XQ, Shi, Y, Muhammad, I, Zhou, MR, Wei, XY, Chen, QQ, Li, WQ, Liu, WT, Chen, KM. Characterization of B-BOX gene family and their expression profiles under hormonal, abiotic and metal stresses in Poaceae plants. BMC Genomics 2019, 20, 27. https://doi.org/10.1186/s12864-018-5336-z
Linghu, B, Song, M, Mu, JX, Huang, SH, An, R, Chen, NN, Xie, CG, Zhu, YT, Guan, ZB, Zhang, YF. Comprehensive analysis of U-box E3 ubiquitin ligases gene family revealed BnPUB18 and BnPUB19 negatively regulated drought tolerance in Brassica napus. Industrial Crops & Products 2023, 200, 116875. https://doi.org/10.1016/j.indcrop.2023.116875
Wu W., Zhu S., Xu L., et al. Genome-wide identification of the Liriodendron chinense WRKY gene family and its diverse roles in response to multiple abiotic stress. BMC Plant Biol. 2022, 22, 25. https://doi.org/10.1186/s12870-021-03371-1

Reviewer 3 Report
Authors present the functional, phylogenetic and structural analysis of BBX gene family in Bambusa pervariabilis × Dendrocalamopsis grandis. Research is properly planned and performed. Obtained results are novel and interesting. Following comments should be addressed before the publication.
1. Lines 133-134, 138, 149, 150, 162-165, 166, 248, 249, 250,266, 270, 273, 274, 525, 527-529, etc ; names of proteins should not be italicized.
2. Table 1- names of genes should be italicized.
3. Section 2.2 and 2.3- analyzed are proteins (not italicize), their phylogenetic similarity and conserved motifs, not genes, correct both paragraphs. For example in line 162 should be not „Seven BBX genes…” but „Seven BBX proteins…”
4. Line 255- correct „eight genes” to „eight proteins”.
5. Figure7; In text are provided results for cold stress, but in the Figure 7 description is the salt stress as In Fig 6. Correct it.
6. Section 2.10 Correct the sentence: „Comparison of amino acid sequences of BBX genes of different species” to „Comparison of amino acid sequences of BBX proteins of different species”.
7. Why fungus A. phaeospermum was used in biotic-stress treatments? Explain it and provide citation of previous research.
8. Line 685 correct „(106 spores/mL)” to „(106 spores/mL)”
9. Line 713- provide citation of the 2-∆∆Ct metod.
10. Line 739- provide names and description of supplement data.
Minor editing of English language required
Author Response
Modification instructions
Dear doctor,
Thank you for your comments concerning our manuscript entitled “Identification and characterization of the BBX gene family in Bambusa pervariabilis × Dendrocalamopsis grandis and their potential role under adverse environmental stresses”.
Those comments are all valuable and very helpful for revising and improving our paper, as well as the important guiding significance to our research. Revised portions are marked in the paper. The main corrections in the paper are as follows:
To question 1: Lines 133-134, 138, 149, 150, 162-165, 166, 248, 249, 250,266, 270, 273, 274, 525, 527-529, etc ; names of proteins should not be italicized.
Answer: We appreciate your important comment. According to the suggestion of the reviewer, we have modified the names of protein italics in lines,
144-147: Concerning the subcellular localization of BBX proteins, all but BDBBX7 and BDBBX20 were found to be localized in the nucleus, with BDBBX6, BDBBX9, BDBBX19, and BDBBX21 appearing in the chloroplast, cell membrane, and mitochondrial sites, in addition to the nucleus.
150: we found only one BDBBX family protein (BDBBX6) containing signal peptides, which amounted to 3.6% of the total gene family.
157-164: To analyze the conserved structural domains of the BBX family, we utilized the online software Pfam, as well as CD-Search in Blast (Figure 1A). Among the 21 BDBBX proteins, only BDBBX6, BDBBX17, and BDBBX19 contained three structural domains simultaneously, accounting for 14.3% of the total number of genes. Of these, six BBX proteins, namely BDBBX2, BDBBX7, BDBBX10, BDBBX12, BDBBX14, and BDBBX21, had one Bbox1_BBX-like domain and one Bbox2_BBX-like domain, whereas four BBX proteins, BDBBX13, BDBBX16, BDBBX18, and BDBBX20, had one Bbox1_BBX-like domain and one CCT domain.
174-178: Seven BBX proteins, namely BDBBX6, BDBBX13, BDBBX16, BDBBX17, BDBBX18, BDBBX19, and BDBBX20, contained motif 2, whereas BDBBX2, BDBBX9, BDBBX10, BDBBX14, BDBBX15, and BDBBX21 were the six BBX proteins containing motif 4. Moreover, only BDBBX6 and BDBBX19 held motif 7.
178: The presence of different motifs among different BDBBX suggests that BBX proteins may have different biochemical characteristics and biological functions.
195: Phylogenetic tree of the neighbor-joining method for the BBX proteins of Bambusa pervariabilis × Dendrocalamopsis grandis, Arabidopsis thaliana, and Oryza sativa.
204-211: Group V was identified as the largest subfamily, which contained 8 BDBBX proteins, representing 38.11% of the total number of gene families. Group I, Group III, and Group IV contained 3, 4, and 6 BDBBX proteins, respectively. The evolutionary tree illustrated that BDBBX was more closely related to OsBBX and distantly related to AtBBX. Within this, OsBBX1 and BDBBX2, OsBBX24 and BDBBX4, OsBBX30 and BDBBX7, OsBBX6 and BDBBX9, OsBBX9 and BDBBX13, OsBBX4 and BDBBX14, and OsBBX11 and BDBBX21 were identified as direct homologous genes.
266: BDBBX1, BDBBX13, and BDBBX17 were predicted to undergo phosphorylation modifications at two protein kinase phosphorylation sites, namely serine and threonine, respectively.
268:BDBBX3 and BDBBX7 were predicted to undergo phosphorylation modifications at two protein kinase phosphorylation sites: serine and tyrosine.
273: and the results indicated that eight proteins in the BDBBX family,
275: Among these proteins,
284: The results revealed that 14 proteins in the BDBBX family were predicted to interact (Figure 4).
288: Expanding the interaction network of other proteins with BBX proteins revealed that BDBBX10, BDBBX11, and BDBBX21 are interactive with SIGE proteins,
290: indicating that BDBBX10, BDBBX11, and BDBBX21 proteins might be involved in this regulation.
291: Moreover, BDBBX8, BDBBX10, BDBBX11, BDBBX13, BDBBX19, and BDBBX21 interact with CCA1 proteins to regulate plant circadian rhythms cooperatively.
293: Furthermore, BDBBX4, BDBBX9, BDBBX10, BDBBX11, BDBBX12, BDBBX15, and BDBBX21 are interactive with the ubiquitin ligase COP1 protein and contribute to photomorphogenesis.
451: Comparison of amino acid sequences of BBX proteins of different species
452-465: The previous experiments indicated that three BDBBX proteins in hybrid bamboo demonstrated resistance against different biotic and abiotic stresses. These proteins, namely BDBBX10, BDBBX15, and BDBBX21, were chosen for multiple sequence alignment, which also included three species of Setaria italica, Oryza sativa, and Panicum virgatum (Figure 9). The outcome revealed that BDBBX10 exhibited 83.33%, 81.60%, and 78.00% similarity to SiBBX22, OsBBX22, and PvBBX22, respectively, with SiBBX22 having the highest similarity. The similarities of BDBBX15 with the three Gramineae species compared exceeded 65.17%. BDBBX21 shared the highest similarity of 81.01% with OsBBX24 and 78.93%, and 77.91% with SiBBX22 and PvBBX22, respectively. In conclusion, BDBBX10, BDBBX15, and BDBBX21 demonstrated substantial homology with the amino acid sequences of Setaria italica, Oryza sativa, and Panicum virgatum. Notably, BDBBX10 and BDBBX21 showed higher similarity to these Gramineae crops, with BDBBX10 sharing the highest similarity to Panicum virgatum and BDBBX21 sharing the highest similarity with Oryza sativa.
469: The multiple sequence alignments of BBX proteins from three different species.
579-580: For example, the N-terminal BBX region of AtBBX32 interacts with the BBX protein of GmBBX62 in Glycine max [44].
582-585: Among these, BDBBX8 interacts with a variety of BDBBX proteins, such as BDBBX10, BDBBX11, BDBBX15, and BDBBX21. Meanwhile, BDBBX10, BDBBX11, and BDBBX21 interact with various other proteins, including the SIGE protein, CCA1 protein, AT5G64170 protein, and COP1 protein.
To question 2: Table 1- names of genes should be italicized.
Answer: We appreciate your important comment. According to the suggestion of the reviewer, we have modified the gene names without italics in Table 1.
Table 1. BBX gene family information of Bambusa pervariabilis × Dendrocalamopsis grandis.
Transcriptome ID |
Gene |
Protein length (aa) |
Molecular weight (Da) |
Theoretical pI |
Instability index |
Aliphatic index |
Grand average of hydropathicity (GRAVY) |
Signal peptide |
Subcellular localization |
PH01000042G1610 |
BDBBX1 |
127 |
13945.96 |
10.26 |
70.44 |
60.08 |
-0.521 |
NO |
Nucleus |
PH01000149G0140 |
BDBBX2 |
227 |
24305.66 |
4.82 |
59.77 |
80.84 |
-0.013 |
NO |
Nucleus |
PH01000192G1360 |
BDBBX3 |
210 |
22874.85 |
11.57 |
63.94 |
90.29 |
-0.337 |
NO |
Nucleus |
PH01000303G0990 |
BDBBX4 |
309 |
32565.29 |
6.41 |
55.53 |
58.25 |
-0.518 |
NO |
Chloroplast.Nucleus |
PH01000450G1030 |
BDBBX5 |
102 |
10539.61 |
8.66 |
74.10 |
51.08 |
-0.52 |
NO |
Chloroplast. Cytoplasm |
PH01000481G0620 |
BDBBX6 |
383 |
41494.94 |
7.59 |
41.16 |
67.44 |
-0.295 |
NO |
Cell membrane.Nucleus |
PH01000616G0330 |
BDBBX7 |
83 |
8412.52 |
6.25 |
56.72 |
71.93 |
-0.014 |
NO |
Nucleus |
PH01000917G0300 |
BDBBX8 |
160 |
16681.41 |
10.50 |
73.87 |
62.62 |
-0.358 |
NO |
Nucleus |
PH01001451G0380 |
BDBBX9 |
143 |
14723.24 |
4.69 |
61.26 |
67.69 |
-0.113 |
NO |
Nucleus |
PH01001725G0310 |
BDBBX10 |
270 |
28776.28 |
7.48 |
50.37 |
73.89 |
-0.087 |
NO |
Nucleus |
PH01001870G0430 |
BDBBX11 |
239 |
26213.84 |
11.48 |
63.61 |
86.65 |
-0.431 |
NO |
Nucleus |
PH01002961G0180 |
BDBBX12 |
90 |
10098.74 |
9.23 |
56.06 |
58.78 |
-0.371 |
NO |
Nucleus |
PH01003421G0140 |
BDBBX13 |
198 |
21263.84 |
5.30 |
52.63 |
64.70 |
-0.366 |
NO |
Nucleus |
Phyllostachys_edulis_newGene_30773 |
BDBBX14 |
241 |
26221.19 |
7.86 |
41.89 |
81.08 |
-0.126 |
NO |
Nucleus |
Phyllostachys_edulis_newGene_38997 |
BDBBX15 |
269 |
29499.68 |
6.45 |
60.16 |
82.75 |
-0.099 |
NO |
Nucleus |
PH01000037G0060 |
BDBBX16 |
194 |
20980.48 |
9.49 |
79.35 |
55.46 |
-0.699 |
NO |
Nucleus |
PH01000780G0510 |
BDBBX17 |
332 |
36625.69 |
11.90 |
93.79 |
41.08 |
-0.932 |
NO |
Chloroplas. |
PH01002146G0170 |
BDBBX18 |
409 |
44318.82 |
6.43 |
63.00 |
58.58 |
-0.593 |
NO |
Nucleus |
PH01002727G0080 |
BDBBX19 |
316 |
34063.68 |
6.08 |
37.30 |
74.56 |
-0.199 |
NO |
Nucleus |
Phyllostachys_edulis_newGene_62371 |
BDBBX20 |
298 |
32384.84 |
6.81 |
59.12 |
74.03 |
-0.345 |
NO |
Nucleus |
PH01003160G0610 |
BDBBX21 |
256 |
27100.62 |
4.96 |
50.25 |
76.33 |
-0.155 |
NO |
Nucleus |
To question 3: Section 2.2 and 2.3- analyzed are proteins (not italicize), their phylogenetic similarity and conserved motifs, not genes, correct both paragraphs. For example in line 162 should be not „Seven BBX genes…” but „Seven BBX proteins…”
Answer: We appreciate your important comment. According to the suggestion of the reviewer, we have modified the analysis in sections 2.2 and 2.3.
2.2 Conserved motif analysis of the BBX gene family
158-163: Among the 21 BDBBX proteins, only BDBBX6, BDBBX17, and BDBBX19 contained three structural domains simultaneously, accounting for 14.3% of the total number of genes. Of these, six BBX proteins, namely BDBBX2, BDBBX7, BDBBX10, BDBBX12, BDBBX14, and BDBBX21, had one Bbox1_BBX-like domain and one Bbox2_BBX-like domain, whereas four BBX proteins, BDBBX13, BDBBX16, BDBBX18, and BDBBX20, had one Bbox1_BBX-like domain and one CCT domain. On the other hand, the remaining eight BDBBX proteins contained only one Bbox1_BBX-like domain.
174-179: Seven BBX proteins, namely BDBBX6, BDBBX13, BDBBX16, BDBBX17, BDBBX18, BDBBX19, and BDBBX20, contained motif 2, whereas BDBBX2, BDBBX9, BDBBX10, BDBBX14, BDBBX15, and BDBBX21 were the six BBX proteins containing motif 4. Moreover, only BDBBX6 and BDBBX19 held motif 7. The presence of different motifs among different BDBBX suggests that BBX proteins may have different biochemical characteristics and biological functions.
2.3 Phylogenetic analysis of the BBX gene family
204-211: Group V was identified as the largest subfamily, which contained 8 BDBBX proteins, representing 38.11% of the total number of gene families. Group I, Group III, and Group IV contained 3, 4, and 6 BDBBX proteins, respectively. The evolutionary tree illustrated that BDBBX was more closely related to OsBBX and distantly related to AtBBX. Within this, OsBBX1 and BDBBX2, OsBBX24 and BDBBX4, OsBBX30 and BDBBX7, OsBBX6 and BDBBX9, OsBBX9 and BDBBX13, OsBBX4 and BDBBX14, and OsBBX11 and BDBBX21 were identified as direct homologous genes.
To question 4: Line 255- correct „eight genes” to „eight proteins”.
Answer: We appreciate your important comment. According to the suggestion of the reviewer, we have revised it to lines 273 “ the results indicated that eight proteins in the BDBBX family,”
To question 5: Figure7; In text are provided results for cold stress, but in the Figure 7 description is the salt stress as In Fig 6. Correct it.
Answer: We appreciate your important comment. According to the suggestion of the reviewer, we have revised it to lines 389-392
“Figure 7. Changes in the expression of 21 BDBBX genes over time in three varieties of hybrid bamboo under cold stress. Histogram showing the changes in the expression of 21 BDBBX genes over time in three varieties of hybrid bamboo under cold stress. Transfer the hybrid bamboo to a cooler (SANYO) at 10 °C for stress. ”
To question 6: Section 2.10 Correct the sentence: „Comparison of amino acid sequences of BBX genes of different species” to „Comparison of amino acid sequences of BBX proteins of different species”.
Answer: We appreciate your important comment. According to the suggestion of the reviewer, we have revised it to lines 451-465
“2.10 Comparison of amino acid sequences of BBX proteins of different species
The previous experiments indicated that three BDBBX proteins in hybrid bamboo demonstrated resistance against different biotic and abiotic stresses. These proteins, namely BDBBX10, BDBBX15, and BDBBX21, were chosen for multiple sequence alignment, which also included three species of Setaria italica, Oryza sativa, and Panicum virgatum (Figure 9). The outcome revealed that BDBBX10 exhibited 83.33%, 81.60%, and 78.00% similarity to SiBBX22, OsBBX22, and PvBBX22, respectively, with SiBBX22 having the highest similarity. The similarities of BDBBX15 with the three Gramineae species compared exceeded 65.17%. BDBBX21 shared the highest similarity of 81.01% with OsBBX24 and 78.93%, and 77.91% with SiBBX22 and PvBBX22, respectively. In conclusion, BDBBX10, BDBBX15, and BDBBX21 demonstrated substantial homology with the amino acid sequences of Setaria italica, Oryza sativa, and Panicum virgatum. Notably, BDBBX10 and BDBBX21 showed higher similarity to these Gramineae crops, with BDBBX10 sharing the highest similarity to Panicum virgatum and BDBBX21 sharing the highest similarity with Oryza sativa. ”
To question 7: Why fungus A. phaeospermum was used in biotic-stress treatments? Explain it and provide citation of previous research.
Answer: We appreciate your important comment. According to the suggestion of the reviewer, we have added the following conclusions in lines 49-50 of the introduction.
“Previous studies have shown that the main pathogen causing shoot blight in hybrid bamboo is Arthrinium phaeospermum [3,9].”,
The reason we chose Arthrinium phaeospermum as a treatment for biotic stress is that Arthrinium phaeospermum (Corda) M.B. Ellis was isolated from diseased tissues of hybrid bamboos in Tianquan County by previous researchers, which was determined to be the pathogen causing tip blight in hybrid bamboos after Koch's law.
[3] Li, S.; Tang, Y.; Fang, X.; et al. Whole-genome sequence of Arthrinium phaeospermum, a globally distributed pathogenic fungus. Genomics, 2020, 112, (1), 919-929.
[9] Li, S.; Fang, X.; Han, S.; et al. Differential proteome analysis of hybrid bamboo (Bambusa pervariabilis × Dendrocalamopsis grandis) under fungal stress (Arthrinium phaeospermum). Scientific Reports, 2019, 9, 18681.
To question 8: Line 685 correct „(106 spores/mL)” to „(106 spores/mL)”
Answer: We appreciate your important comment. According to the suggestion of the reviewer, we have revised it to lines 753 “The spores were then diluted using sterile saline to prepare a suspension of A. phaeospermum spores (106 spores/mL).”
To question 9: Line 713- provide citation of the 2-∆∆Ct metod.
Answer: We appreciate your important comment. According to the suggestion of the reviewer, we have revised it to lines 794 “To minimize errors, three replicates of each experiment were conducted. Relative transcript abundance values were computed through the 2-∆∆Ct method [52].”
- 52. Livak, J.;Schmittgen, T.D.Analysis of relative gene expression data using realtime quantitative PCR and the 2(-Delta Delta C(T)) Method. Methods 2001, 25, (4), 402-408. doi:10.1006/meth.2001.1262
To question 10: Line 739- provide names and description of supplement data.
Answer: We appreciate your important comment. According to the suggestion of the reviewer, we have revised it to lines 821-825
“Supplementary Materials: Table S1. qPCR primers for 21 BDBBX genes. Table S2. Disease grading criteria. Table S3. Results of disease index and incidence rate of three B. pervariabilis × D. grandis varieties. Figure S1. Potential phosphorylation sites for the BDBBX gene family. Figure S2. (A) Disease index of three B. pervariabilis × D. grandis varieties; (B) Correlation analysis of gene expression with incidence rate (Ir) and Disease index (Di). * indicates p value less than 0.05, ** indicates p value less than 0.01, and *** indicates p value less than 0.001.”
To question: Minor editing of English language required.
Answer: We appreciate your important comment. According to your suggestion, we have invited native language experts to help revise this article.

Round 2
Reviewer 2 Report
Dear Authors
The authors provided fairly detailed responses to my comments and made some changes to the text of the manuscript. We would like to hope that in the future, the level of transcripts of certain genes will not be directly associated with resistance without its direct assessment. References to the fact that in other publications often understand PEG-induced stress as drought are not convincing, since PEG causes osmotic stress, which initiates water deficiency in the plant. Drought as a stress factor includes not only water scarcity, but also other damaging factors.
I recommend accepting the manuscript in the submitted form.
Kind regards
Reviewer 3 Report
Manuscript is substantially improved, I have no comments.